# Spatiotemporal Distribution and Risk Assessment of Heat Waves Based on Apparent Temperature in the One Belt and One Road Region

**Cong Yin** [1,2], **Fei Yang** [1,3,*], **Juanle Wang** [1] **and Yexing Ye** [1,2]

1   State Key Laboratory of Resources and Environmental Information System, Institute of Geographic Sciences and Natural Resources Research, Chinese Academy of Sciences, Beijing 100101, China; yinc.18s@igsnrr.ac.cn (C.Y.); wangjl@igsnrr.ac.cn (J.W.); yeyx.17s@igsnrr.ac.cn (Y.Y.)
2   University of Chinese Academy of Sciences, Beijing 100049, China
3   Jiangsu Center for Collaborative Innovation in Geographical Information Resource Development and Application, Nanjing 210023, China
*   Correspondence: yangfei@igsnrr.ac.cn

**Abstract:** Heat waves seriously affect the productivity and daily life of human beings. Therefore, they bring great risks and uncertainties for the further development of countries in the One Belt and One Road (OBOR) region. In this study, we used daily meteorological monitoring data to calculate the daily apparent temperature and annual heat wave dataset for 1989–2018 in the OBOR region. Then, we studied their spatiotemporal distribution patterns. Additionally, multi-source data were used to assess heat wave risk in the OBOR region. The main results are as follows: (1) The daily apparent temperature dataset and annual heat wave dataset for 1989–2018 in the OBOR region at $0.1° \times 0.1°$ gridded resolution were calculated. China, South Asia and Southeast Asia are suffering the most serious heat waves in the OBOR region, with an average of more than six heat waves, lasting for more than 60 days and the extreme apparent temperature has reached over 40 °C. Additionally, the frequency, duration and intensity of heat waves have been confirmed to increase continuously. (2) The heat wave risk in the OBOR region was assessed. Results show that the high heat wave risk areas are distributed in eastern China, northern South Asia and some cities. The main conclusion is that the heat wave risk in most areas along the OBOR route is relatively high. In the process of deepening the development of countries in the OBOR region, heat wave risk should be fully considered.

**Keywords:** One Belt and One Road; heat wave; apparent temperature; spatiotemporal distribution; risk assessment

## 1. Introduction

In the Intergovernmental Panel on Climate Change (IPCC) special report, human activities are estimated to have caused approximately 1.0 °C of global warming above pre-industrial levels, with a likely range of 0.8 °C to 1.2 °C (https://www.ipcc.ch/sr15/). Global warming is projected to intensify heat wave events as quantified by multiple descriptors, including frequency, duration and intensity [1,2]. The OBOR region involves 3 continents, more than 66 countries and regions and approximately 4.4 billion people, with frequent natural disasters, highly concentrated populations, and fragile ecological environments [3,4]. From 1995 to 2015, among the 10 most severe meteorological disaster-affected countries, 7 are in the OBOR region. In addition, most countries in this region are developing countries, which have poor resilience to natural disasters [5]. In recent years, extreme heat wave events have occurred frequently in many parts of the OBOR region, causing serious casualties and property

losses. For example, in 2010, a heat wave lasted for three weeks in Russia and killed approximately 56,000 people c (https://www.reuters.com/article/us-russia-heat-deaths-idUSTRE69O4LB20101025). From April to June 2015, India experienced a bout of intense heat waves that killed more than 2500 people across the country. During the same period, a heat wave in southern Pakistan killed more than 800 people [6].

The direct adverse effects of heat waves include: (1) for human health, cardiovascular and respiratory damage, even death [7–12]; (2) during heat waves, demand for water and electricity surges, putting pressure on water and electricity supplies [13–17]; (3) heat waves affect labor efficiency and reduce productivity [18–21]; (4) heat waves cause drought, which affects crop growth and reduces crop yield [22–24]; (5) heat waves affect the growth and reproduction of livestock and poultry, further affecting the production of meat, eggs and milk [25–28]. Studying heat waves in the OBOR region can provide a basis for disaster risk reduction, infrastructure construction and enterprise investment and effectively help the government, residents and enterprises in disaster risk reduction and prevention, which are of great significance for the further development of countries in the OBOR region.

Research on heat waves has focused on the following aspects: (1) heat wave data production [29,30]; (2) the impact of heat waves on human casualties, productivity and infrastructure, etc. [9]; (3) heat wave risk assessment [13,31]; (4) the mechanism that caused the heat wave [32–34]; (5) synergy between urban heat islands and heat waves [35–37]; (6) heat wave disaster management mechanisms [38,39]; (7) heat wave disaster prediction [40,41]. Heat wave data production is the basic work of heat wave studies. Raei [29] proposed a multi-method Global Heat Wave and Warm-Spell Data Record and Analysis Toolbox (GHWR). This toolbox was used to create a multi-method global heat wave and warm-spell record based on gridded $0.5° × 0.5°$ temperature data from 1979 to 2017. Malcolm N [42] introduced a new high-resolution global gridded dataset of Climate Extreme Indices (CEIs) based on sub-daily temperature and precipitation data from the Global Land Data Assimilation System (GLDAS). The dataset called "CEI_0p25_1970_2016" includes 71 annual (and in some cases monthly) CEIs at $0.25° × 0.25°$ gridded resolution, covering 47 years over the period 1970–2016 [42]. By simulating daily maximum and minimum temperature and precipitation, Defrance [43] conducted prediction studies and produced a dataset of global extreme climatic indices annually and globally with a resolution of $0.5° × 0.5°$ from 1951 to 2099. The existing data production methods have the following shortcomings: (1) They have more restrictions on the data source (such as the format of the data). At present, the data sources of heat wave data production mainly include meteorological monitoring data and grid data. Meteorological monitoring data are sensitive to missing values and can only be calculated based on discrete monitoring stations, so it is difficult to conduct a global calculation. The data source used by GHWR was CPC (Climate Prediction Center) global daily temperature data, which were organized into NetCDF format from grid data. Therefore, this method is difficult for processing meteorological monitoring data and general grid data. (2) There are other restrictions on data sources. The spatial resolution of CPC global daily temperature data used by GHWR is $0.5°$, and no parameter interface is provided for other resolution data, which also limits the accuracy of the heat wave dataset.

Heat waves have seriously affected the productivity and daily life of human beings in the OBOR region, especially in South and Southeast Asia [6,44]. In order to assess the heat wave risk in this area on a more precise level, we constructed a heat wave dataset and evaluated heat wave risk in the OBOR region using multi-source data. Only using air temperature makes it difficult to reflect the actual experience of people. In addition to air temperature, the actual experience of people is affected by other factors such as wind speed and relative humidity. Taking people's actual experience (air temperature, wind speed and relative humidity) into account, this study used apparent temperature to calculate the annual heat wave dataset for 1989–2018 in the OBOR region at a $0.1° × 0.1°$ gridded resolution. The dataset is now available on IKCEST (International Knowledge Center for Engineering Sciences and Technology). The main contributions of this study are as follows:

- A new heat wave calculation method, CHWT (Combined Heat Wave Threshold), is proposed, which considers the heterogeneity of heat wave thresholds temporally and spatially.

- The annual heat wave dataset for 1989–2018 in the OBOR region was calculated based on CHWT. The heat wave dataset includes Heat Waves Frequency (HWF), Heat Waves Total Duration (HWTD), Heat Waves Maximum Duration (HWMD), Heat Waves Maximum Apparent Temperature (HWMAT), Heat Waves Start Date (HWSD) and Heat Waves End Date (HWED).
- The spatiotemporal distribution of apparent temperature and heat waves in the OBOR region was identified.
- The heat wave risk of the OBOR region was assessed. We point out the high heat wave risk area, which is of great significance for residents life, enterprise investment and tourism planning.

As a lot of abbreviations are used in this article, in order to facilitate reading and understanding, we provide a comparison table of abbreviations and their full names, and briefly explain each abbreviation, as shown in Table 1:

**Table 1.** Abbreviations and their full names and definitions.

| Abbreviation | Full Name | Definition |
|---|---|---|
| HWF | Heat waves frequency | Number of heat waves in a year |
| HWTD | Heat waves total duration | Total duration of heat waves in a year |
| HWMD | Heat waves maximum duration | Duration of the longest heat wave in a year |
| HWMAT | Heat waves maximum apparent temperature | The highest apparent temperature of each heat wave in a year |
| HWSD | Heat waves start date | Start date of the first heat wave in a year |
| HWED | Heat waves end date | End date of the last heat wave in a year |
| RTT | Relative temperature threshold | \ |
| ATT | Absolute temperature threshold | \ |
| CRTT | Climatological relative temperature threshold | Percentile threshold based on historical temperature series for a day |
| ARTT | Annual relative temperature threshold | Percentile threshold based on annual temperature series |
| DT | Duration threshold | \ |

## 2. Materials and Methods

### 2.1. Materials

Meteorological monitoring data and elevation data are used to calculate the apparent temperature dataset and the heat wave dataset. In this study, we used daily meteorological monitoring data from 2833 monitoring stations in the OBOR region [45]. Missing values in meteorological station data must be processed before they can be used for subsequent calculation. Compared with the mean value method and the regression method, the interpolation results of the Random Forest (RF) method are more accurate and reliable [46]. Based on the RF method [47], we used MissForest to process the missing values [48]. Additionally, due to the vertical zonation of temperature, we used the elevation data as an important basis for interpolating daily meteorological monitoring data into grid data [49]. Figure 1a shows the distribution of monitoring stations.

Multi-source data are used to assess heat wave risk in the OBOR region. Definitions of risk are commonly probabilistic in nature, referring to the potential losses from a particular hazard to a specified element at risk in a particular future time period [50,51]. Heat wave risk is the probability of harmful consequences or likelihood of losses resulting from interactions among heat wave hazard (i.e., the possible future occurrence of heat wave events), heat wave exposure (i.e., the total population, its livelihoods and assets in an area in which heat wave events may occur) and heat wave vulnerability (i.e., the propensity of exposed elements to suffer adverse effects when impacted by a heat wave event) [52]. According to the risk triangle evaluation theory [53], we chose the following hazard, exposure and vulnerability factors, as shown in Table 2.

**Table 2.** Hazard, exposure and vulnerability factors.

| Factor | Data | Resolution | Time | Format |
|---|---|---|---|---|
| Hazard | Heat wave frequency | Annually, 0.1° | 1989–2018 | Grid |
| | Heat wave duration | Annually, 0.1° | 1989–2018 | Grid |
| | Heat wave intensity | Annually, 0.1° | 1989–2018 | Grid |
| Exposure | DMSP night-time lights data | Annually, 30″ | 2010 | Grid |
| | CIESIN population data | 30″ | 2010 | Grid |
| Vulnerability | FAO water areas data | / | 2012 | Vector |
| | OSM hospital distribution data | Daily | October, 2019 | Vector |
| | AVHRR NDVI data | Daily, 0.05° | July 1, 2010 | Grid |
| | DRYAD GDP data | Five years, 10 km | 2015 | Grid |

Heat wave hazard is a measure of the severity of heat wave events. We use the frequency, duration and intensity of the heat wave to represent the heat wave hazard. Heat wave exposure is the degree to which people, livelihoods and the economy etc., may be adversely affected. The night-time lights data is used to represent the economic exposure, and population density to represent population exposure [54,55]. The proportion of people under 14 and the proportion of people over 65 constitute the vulnerable groups. Heat wave vulnerability is a measure of the factors that increase/decrease risk in the environment. We use the distance from water/hospital, NDVI and GDP to represent heat wave vulnerability, which can decrease heat wave risk [56–58]. Figure 1b shows the hospital distribution of the OBOR region.

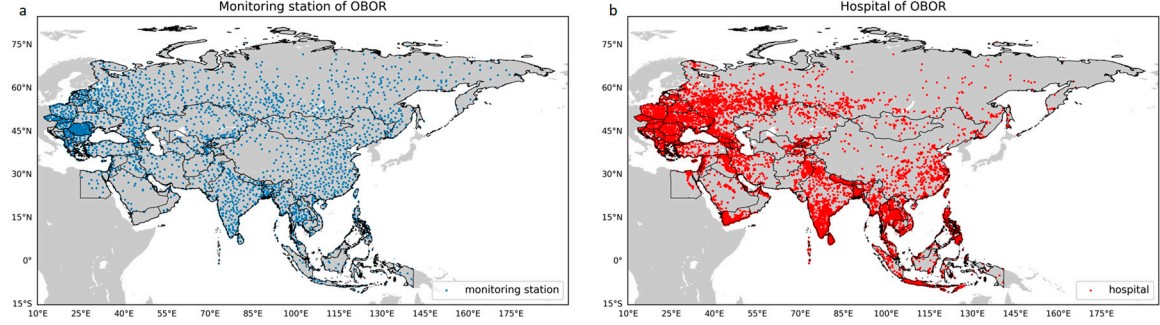

**Figure 1.** Materials used in this study: (**a**) NCEI monitoring station distribution; (**b**) OSM hospital distribution.

*2.2. Methods*

2.2.1. MissForest: Non-Parametric Missing Value Imputation

MissForest is an iterative imputation method based on RF, which can deal with different types of variables (continuous or discrete) at the same time. MissForest does not need to adjust parameters, and does not need assumptions about data distribution [48]. Compared with KNNImpute, MissPALasso and MICE, MissForest has better imputation performance and calculation efficiency [48,59–61]. In this study, we used R package missForest to process missing value of meteorological monitoring data (https://cran.r-project.org/web/packages/missForest/). The calculation method is as follows:

- We assume $X = (X_1, X_2, \ldots, X_p)$ to be a n × p-dimensional data matrix. For an arbitrary variable $X_s$ with missing values, we can separate the dataset into four parts: $y^{(s)}_{obs}$: the observed values of variable $X_s$; $y^{(s)}_{mis}$: the missing values of variable $X_s$; $x^{(s)}_{obs}$: the variables other than $X_s$ with observations; $x^{(s)}_{mis}$: the variables other than missing values of $X_s$ with observations.
- First, an initial guess for the missing values is made in X using the mean/median imputation method. Second, the variables are sorted $X_s$, s=1, ..., p according to the amount of missing values from small to large. For each variable $X_s$, the missing values are imputed by first fitting an RF with response $y^{(s)}_{obs}$ and predictors $x^{(s)}_{obs}$; then, predicting the missing values $y^{(s)}_{mis}$ by applying

the trained RF to x$^{(s)}$$_{mis}$. The imputation procedure is repeated until the latest imputation result is not better than the previous one [48].

### 2.2.2. Elevation-Based Interpolation Method

Because monitoring stations are discretely distributed in the study area, interpolation is needed to obtain data covering the whole study area for the subsequent heat wave calculation. Due to the large scope of the study area, the vertical zonation of temperature must be considered, and the traditional methods such as Inverse Distance Weighting (IDW) and Ordinary Kriging (OK) cannot be used directly for interpolation. Therefore, we adopted an elevation-based interpolation method [62–65]. According to this method, the temperature decreases linearly (slope = 0.0065 °C/m) as the elevation increases. The core idea is as follows:

- First, an initial correction of the temperature is made. The temperature at the given station is corrected to the zero plane. At this time, the temperature has no relationship with elevation but rather only the spatial correlation and spatial heterogeneity exist on the same plane, qualifying the use of a traditional interpolation method (such as Ordinary Kriging), defined as Equation (1):

$$T_{c1} = T_a + 0.0065 \times E \tag{1}$$

  where $T_{c1}$ is first corrected temperature (°C), $T_a$ is the monitoring station temperature (°C), and E is the monitoring station elevation (m).

- Then, the corrected temperature is interpolated to a synthetic mesh of 0.1° x 0.1°. Based on the first corrected temperature, the traditional interpolation method is used for interpolation. In this study, we use Kriging tool in arcpy software package provided by ArcGIS for interpolation, which uses the Ordinary Kriging method, spherical semi variance model and lag size is 0.371455. At this time, we obtained the temperature data covering the whole study area on the assumption of a zero plane, defined as Equation (2):

$$T_{c1} \xrightarrow{\text{Ordinary Kriging}} T_i \tag{2}$$

  where $T_i$ is interpolated temperature (°C).

- Finally, the second temperature correction is performed. Using elevation data, the interpolated temperature data is corrected again to its actual elevation. At this time, the temperature data take topographic features into account, which can show the obvious vertical zonation, defined as Equation (3):

$$T_{c2} = T_i - 0.0065 \times E \tag{3}$$

  where $T_{c2}$ is the second corrected temperature (°C).

### 2.2.3. Apparent Temperature

Heat waves directly cause discomfort and affect human health. The impact of heat waves on human health has been widely studied and reported [9,12,41]. The cold and hot feeling to the external environment of the human body is affected by the comprehensive influence of air temperature, wind speed, relative humidity and solar radiation [66,67]. Based on the existing calculation methods of apparent temperature and the accessibility of data, this study mainly considers the first three factors. As apparent temperature takes into account more environmental factors than air temperature, it can more accurately reflect the cold and hot degree of the external environment [68]. Different from most of the previous studies, which use air temperature to define heat wave, this study uses apparent temperature to define heat wave based on the real feeling of human body to the external environment. We used two methods to calculate the daily apparent temperature of the study area in 1989–2018:

- Humidex index

The Humidex index [69] is increasingly widely used in indoor and outdoor thermal environment evaluations and human comfort evaluation due to its simple calculation and strong explanatory ability. According to the current international standard, the Humidex index has a good evaluation effect on thermal comfort under the condition of high temperature in summer [66,70]. The Humidex index takes air temperature and dew point temperature into account, and its calculation method is as follows:

$$AT = T_a + 0.5555 \times \left( 6.11 \times e^{5417.753 \times \left( \frac{1}{273.16} - \frac{1}{T_d + 273.15} \right)} - 10 \right) \tag{4}$$

where $AT$ is apparent temperature (°C), $T_a$ is air temperature (°C), and $T_d$ is dew point temperature (°C).

- Steadman index

Steadman [68] published the general formula for apparent temperature in 1984. The Steadman index takes three indicators into account: air temperature, water vapor pressure and wind speed. The calculation method is as follows [71]:

$$AT = 1.07 \times T_a + 0.2 \times P - 0.65 \times V - 2.7 \tag{5}$$

$$P = \frac{R_h}{100} \times 6.105 \times e^{\frac{17.27 \times T_a}{237.7 + T_a}} \tag{6}$$

$$R_h = \frac{T_d + 19.2 - 0.84 \times T_a}{0.198 + 0.0017 \times T_a} \tag{7}$$

where $P$ is vapor pressure (Pa), $V$ is wind speed (m/s), and $R_h$ is relative humidity.

### 2.2.4. CHWT: Combined Heat Wave Threshold

Due to the large study area, different regions have different heat wave standards [72–74]. For example, the World Meteorological Organization (WMO) recommends high temperature days when the temperature exceeds 32 °C, and a high temperature weather process that lasts for more than three days is called a heat wave. [12]. In most parts of China, the standard for high temperature days is 35 °C [40]. Therefore, it is unreasonable to use the same temperature threshold to judge heat waves in the whole region. In CHWT, we use the combination of Relative Temperature Threshold (RTT) and Absolute Temperature Threshold (ATT) to define heat wave:

- Climatological Relative Temperature Threshold (CRTT)

When the temperature of a place is higher than the historical temperature for a long time, it reflects the possibility of extreme high temperature. Therefore, for the selected date, we rank the apparent temperature of each grid point in 1989–2018, and then select the temperature corresponding to different percentiles as the RTT to judge heat waves, which is defined as the Climatological Relative Temperature Threshold (CRTT). CHWT allows setting different percentile thresholds to accommodate different heat wave standards. In this study, by referring to other work and comparing the results of different thresholds (CRTT = 85 and 90), we found that CRTT = 90 best reflects the actual state of heat waves [29,75].

- Annual Relative Temperature Threshold (ARTT)

When the temperature of a certain day is a higher value in the temperature series of that year, it also reflects the possibility of extreme high temperature. Therefore, we rank the daily apparent temperature of each grid point every year, and define the RTT by setting different percentile thresholds, which is defined as Annual Relative Temperature Threshold (ARTT). In this study, we calculated the heat wave dataset with ARTT = 80 and ARTT = 85.

- Absolute Temperature Threshold (ATT)

When the temperature is higher than RTT, it does not necessarily mean the occurrence of heat waves (such as winter). Therefore, we also set an absolute temperature threshold to avoid this situation.

In this study, we use different combinations of CRTT and ATT, ARTT and ATT to define the high temperature threshold. The weather process that reaches the high temperature threshold and Duration Threshold (DT) is called a heat wave.

## 3. Results

### 3.1. Dataset Validation

Accurate apparent temperatures are the premise of heat wave calculation. Therefore, we use a cross validation method to verify the apparent temperature dataset, and evaluate the accuracy performance of the elevation-based interpolation method by comparing the interpolation results with the calculation results based on the station monitoring values. The specific method is as follows:

- 15% of the daily available monitoring stations are randomly selected as the verification set, which are discretely distributed in the whole study area at different altitudes. The apparent temperature of each station in the verification set is calculated as the real value.
- Using the elevation-based interpolation method, the daily apparent temperature grid data covering the whole study area are obtained from all available monitoring stations.
- The apparent temperature at the validation set stations are extracted from the daily apparent temperature grid data as the predictive value.
- By comparing the predictive value with the corresponding real value, the validation result is obtained.

In this study, we verified the whole time series of the apparent temperature dataset and we reported the results of each cross validation in the form of a table. The results show that the minimum slope is 0.9695 and the maximum is 1.0206, and the minimum Coefficient of Determination ($R^2$) is 0.9767, showing that predictive value and true value are highly correlated. The maximum P value is $9 \times e^{-323}$, indicating that the results of linear regression are significant. The maximum Mean Absolute Error (MAE) is 1.849, and the maximum Root Mean Square Error (RMSE) is 2.7362. In summary, the validation result shows that the elevation-based interpolation method is accurate and reliable. Thus, the apparent temperature dataset can be used for heat wave calculation. Additionally, in order to visualize those areas where the interpolation method has lower performance, by averaging the prediction variance map of the entire time series, Figure 2 shows the uncertainty map based on the elevation-based interpolation method. The smaller the variance, the closer the results of multiple interpolations are, and the smaller the uncertainty is.

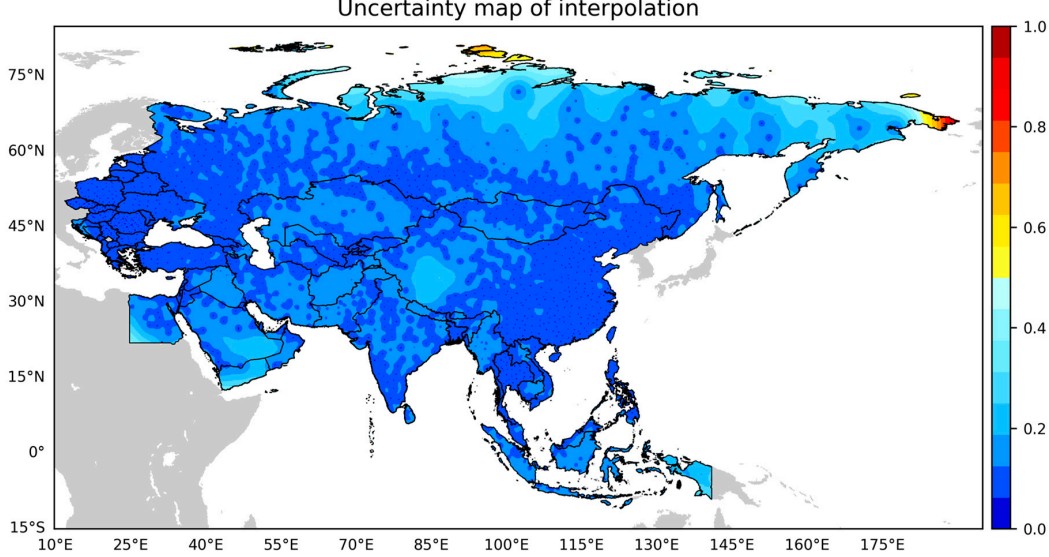

**Figure 2.** Uncertainty map of interpolation.

Figure 3 and Table 3 show six cross validation results: the slope is higher than 0.99 and lower than 1.01, the $R^2$ is higher than 0.96, and the result passed the significance test (P value < 0.01). Using maximum-minimum normalization method, Table 3 also shows the normalized mean absolute error (NMAE) and normalized root mean square error (NRMSE). In Figure 3, we find some points that deviate more from the straight line, which is mainly caused by some monitoring stations located in special locations (such as a valley mouth with high wind speed, a water area with high relative humidity), which will result in the calculated apparent temperature being higher or lower. Our interpolation method shields the local climate to a certain extent, thus reflecting the overall apparent temperature level.

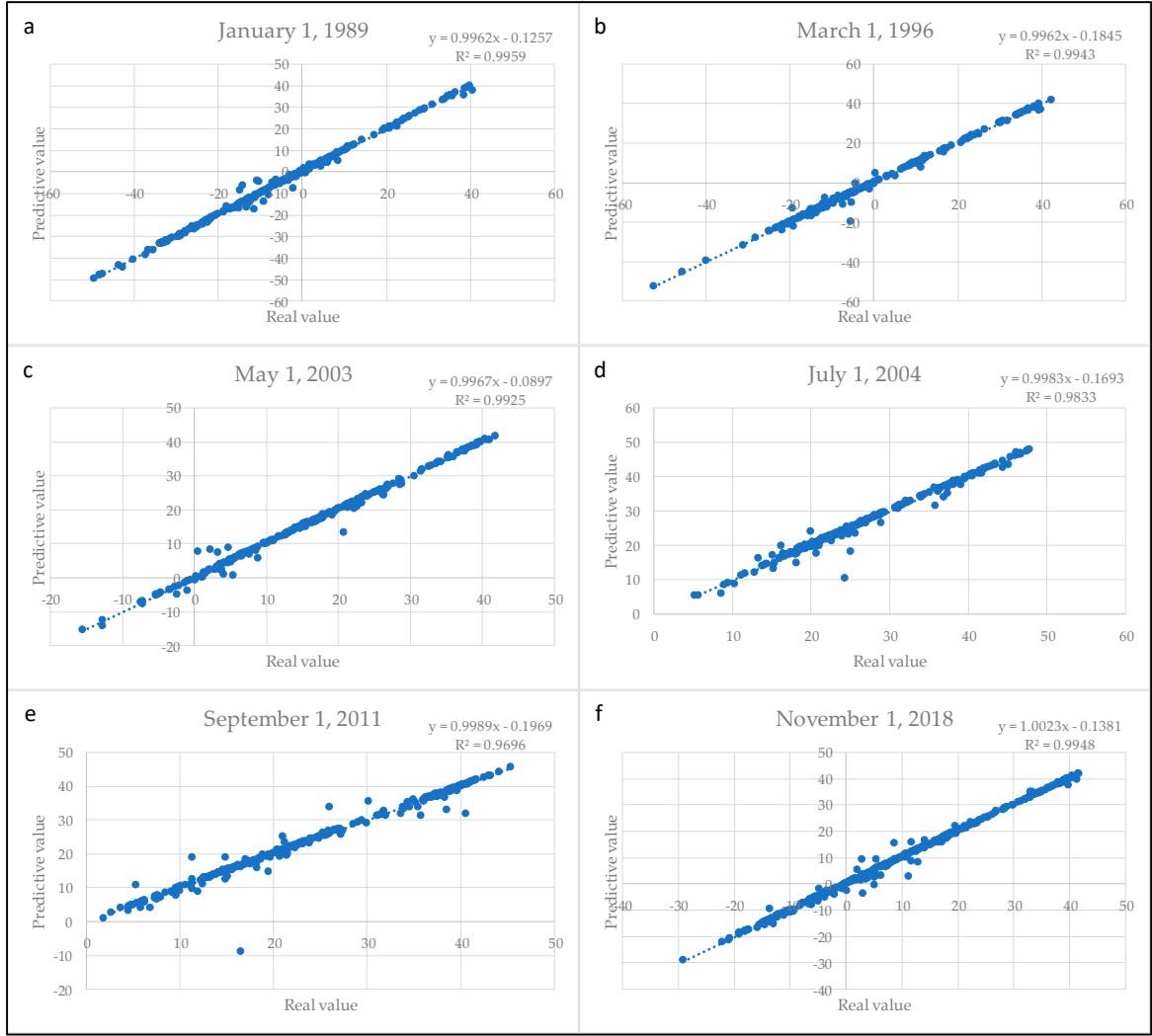

**Figure 3.** Validation result of apparent temperature for: (**a**) January 1, 1989; (**b**) March 1, 1996; (**c**) May 1, 2003; (**d**) July 1, 2004; (**e**) September 1, 2011; (**f**) November 1, 2018.

In 2010, due to the occurrence of the Russian heat wave, the frequency, duration and intensity of the heat waves in the OBOR region were significantly higher than those of other years [29,76]. To verify the performance of the heat wave dataset, we evaluated the availability of the dataset during the 2010 heat wave. Figure 4 shows the comparison of HWF, HWTD and HWMAT under two thresholds. In the two results, the region of heat wave occurrence is basically the same, the Russian heat wave in summer 2010 is successfully detected (Figure 4a,b), and the spatial distribution of HWMAT is almost the same (Figure 4e,f). Due to the different definition methods of temperature threshold, the results based on CRTT have obvious spatial heterogeneity, while the results based on ARTT have obvious

spatial correlation, which show the high value area (top 20% of the value series, same as below) in the former is discrete and dotted distribution (Figure 4a,c), while the high value area in the latter is continuous and contiguous distribution (Figure 4b,d). In this calculation result, the high value area of HWF is North India and North Southeast Asia (Figure 4b), the high value area of HWTD is southeast China, North Southeast Asia, South Asia, South Central Asia and the Arabian Peninsula (Figure 4d), and the high value areas of HWMAT are eastern China and northern India (Figure 4f), which are consistent with other studies [29,44,77]. Therefore, the calculation results based on ARTT can better reflect the actual spatial distribution of heat wave.

**Table 3.** Validation result.

| Date | Slope | $R^2$ | P Value | MAE | NMAE | RMSE | NRMSE |
|---|---|---|---|---|---|---|---|
| January 1, 1989 | 0.9962 | 0.9959 | $2.994 \times e^{-321}$ | 0.527 | 0.3166 | 1.2028 | 0.108 |
| March 1, 1996 | 0.9962 | 0.9943 | 0 | 0.5256 | 0.3087 | 1.3764 | 0.3027 |
| May 1, 2003 | 0.9967 | 0.9925 | 0 | 0.4709 | 0 | 1.1065 | 0 |
| July 1, 2004 | 0.9983 | 0.9833 | 0 | 0.4917 | 0.1174 | 1.2875 | 0.203 |
| September 1, 2011 | 0.9989 | 0.9696 | $1.0533 \times e^{-312}$ | 0.6481 | 1 | 1.9982 | 1 |
| November 1, 2018 | 1.0023 | 0.9948 | 0 | 0.5922 | 0.6845 | 1.1411 | 0.0388 |

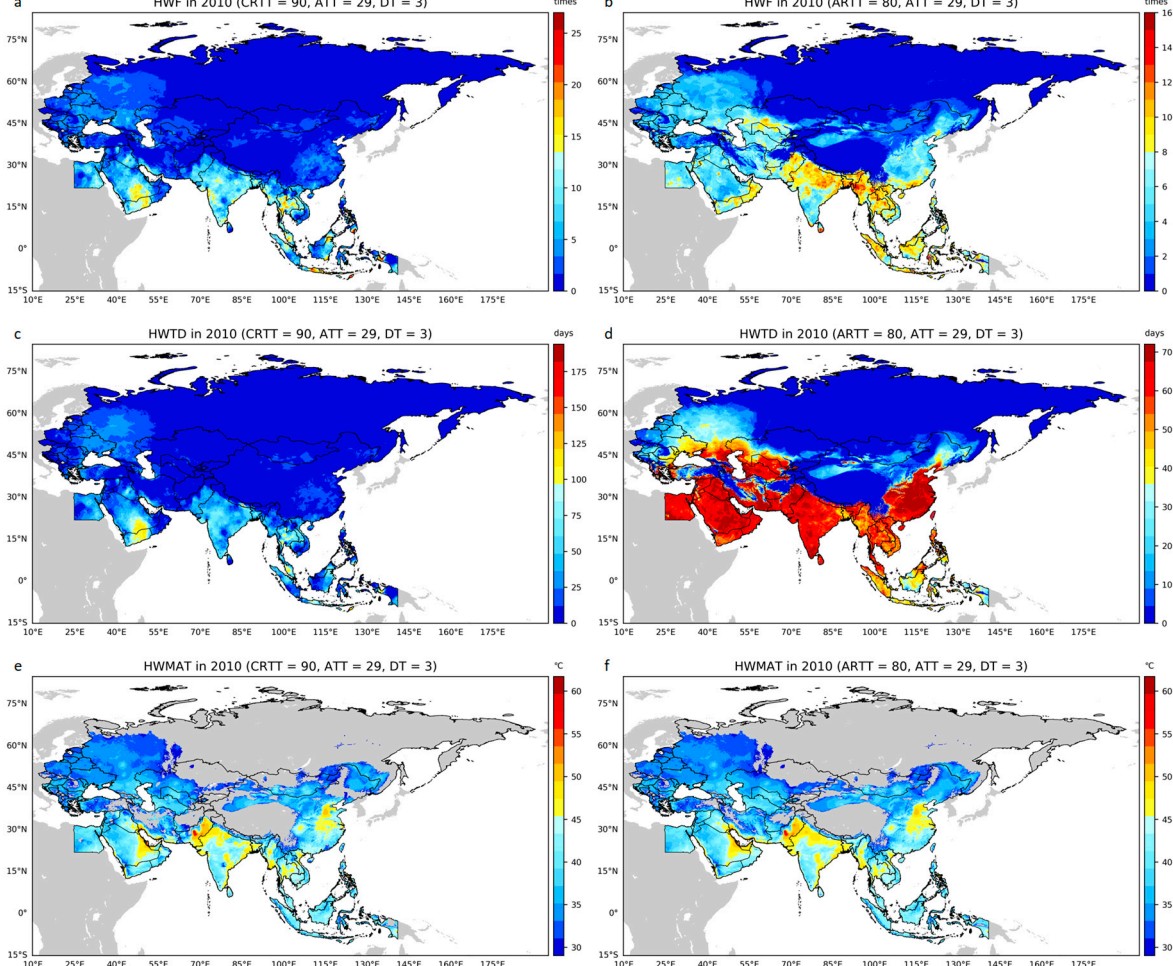

**Figure 4.** Comparison of the results under two thresholds: (**a,c,e**) HWF, HWTD and HWMAT under CRTT = 90; (**b,d,f**) HWF, HWTD and HWMAT under ARTT = 80.

### 3.2. Seasonal Variation of Apparent Temperature for 1989–2018

Spatially, using the Humidex index, Figure 5 shows the mean apparent temperature of four seasons over 30 years. With the movement of the direct sunlight region, the high temperature region began to expand northward in spring (Figure 5a), and its area reached the maximum in summer and began to contract southward (Figure 5b), and continued to contract southward in autumn and reached the minimum in winter (Figure 5c,d). In spring, the average apparent temperature of South Asia, Southeast Asia and parts of the Arabian Peninsula in low latitude (15° S-30° N) has reached over 29 °C. Affected by the high relative humidity, the high value areas appear in the coastal areas (Figure 5a). In summer, the average apparent temperature in East China and South-Central Asia also reached over 29 °C. Affected by the direct sunlight, the extremely high value area appears near the Tropic of Cancer (Figure 5b). The distribution of the high temperature area in autumn is consistent with that in spring (Figure 5c). In winter, the area above 29 °C is limited to 15° S-20° N, and the highest value appears near the equator (Figure 5d).

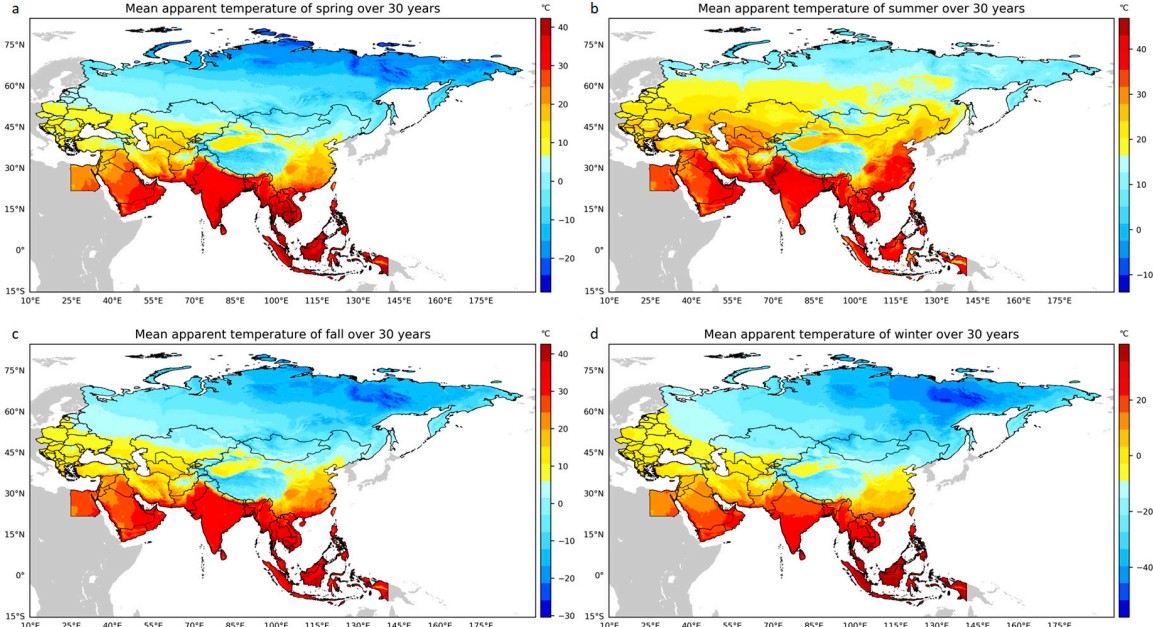

**Figure 5.** Spatial distribution of mean apparent temperature between 1989–2018 for: (**a**) spring, (**b**) summer, (**c**) autumn, (**d**) winter.

We take the time series (1989–2018) as the independent variable, take the average apparent temperature of each season in 30 years as the dependent variable, carry out the linear regression between the two variables, and calculate the change slope of the apparent temperature of each season in 30 years. The significance test was performed. Figure 6 shows the slope of the mean apparent temperature of four seasons over 30 years of the regions passing the significance test (P value < 0.05). A slope higher than 0 indicates that apparent temperature has a linear increasing trend; a slope lower than 0 indicates that apparent temperature has a linear decreasing trend. For the regions passing the significance test, 98.7%, 96.7%, 98.2% and 83.9% of the four seasons have slopes greater than 0, which are higher than the proportion without significant tests (92.5%, 90.1%, 89.9% and 56.5%), which indicates that the apparent temperature in most regions has a rising trend in the past 30 years, and the trend of the increase in apparent temperature is universal and significant, while the trend of the decrease in apparent temperature is limited and not significant. In spring and summer, the high value area is inland, and the distribution is relatively discrete. The slope of coastal area is close to 0, indicating that the apparent temperature is not significantly increased (Figure 6a,b). In autumn and winter, the high value area is coastal area, which is continuous and layered in distribution. The slope of

inland area is less than 0, indicating the apparent temperature tends to decrease. As a sensitive region of climate change [78,79], the slope of the Tibetan Plateau and the Arctic region has reached more than 0.1, indicating that the apparent temperature has increased dramatically (Figure 6c,d).

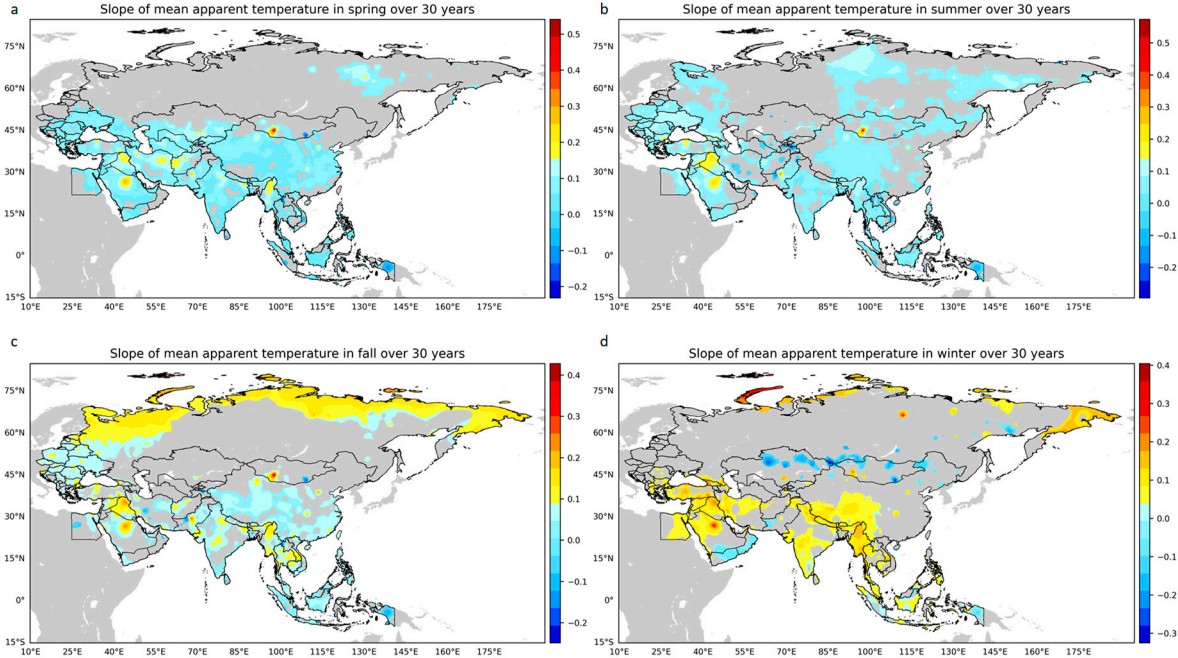

**Figure 6.** Slope of mean apparent temperature for: (**a**) spring, (**b**) summer, (**c**) autumn, (**d**) winter.

### 3.3. Spatiotemporal Distribution of Heat Wave for 1989–2018

Based on ARTT = 80, ATT = 29 and DT = 3, Figure 7 shows the annual average value of HWF, HWTD and other six heat wave attributes in the OBOR region from 1989 to 2018. In terms of HWMAT, HWSD and HWED, only areas with consecutive heat waves in 30 years are calculated. Southeast Asia and India suffered the most frequent heat waves, with an average of more than 6 times. Among them, the frequency of heat waves in the north of Southeast Asia and the northeast of India reached more than 9 times. The average number of heat waves in eastern China, Central Asia and Western Asia is 4–6 (Figure 7a). Heat waves in eastern China, India and the Arabian Peninsula last the longest, reaching more than 60 days (Figure 7b). The duration of a single heat wave in the coastal areas of eastern China, western India and the Arabian Peninsula is longer, reaching more than 30 days, while the duration of a single heat wave in southern Southeast Asia is shorter, less than 10 days (Figure 7c). The extreme apparent temperature occurs along the India-Pakistan border and the Persian Gulf Coast, and the apparent temperature exceeds 50 °C. The apparent temperature of eastern China, northern India and Northern Southeast Asia also reached over 40 °C (Figure 7d). According to Figure 7e,f, the first heat wave in eastern China and the Arabian Peninsula starts from day 150–175, the first heat wave in India starts from day 100–125, and the last heat wave in these three regions ends from day 225–250.

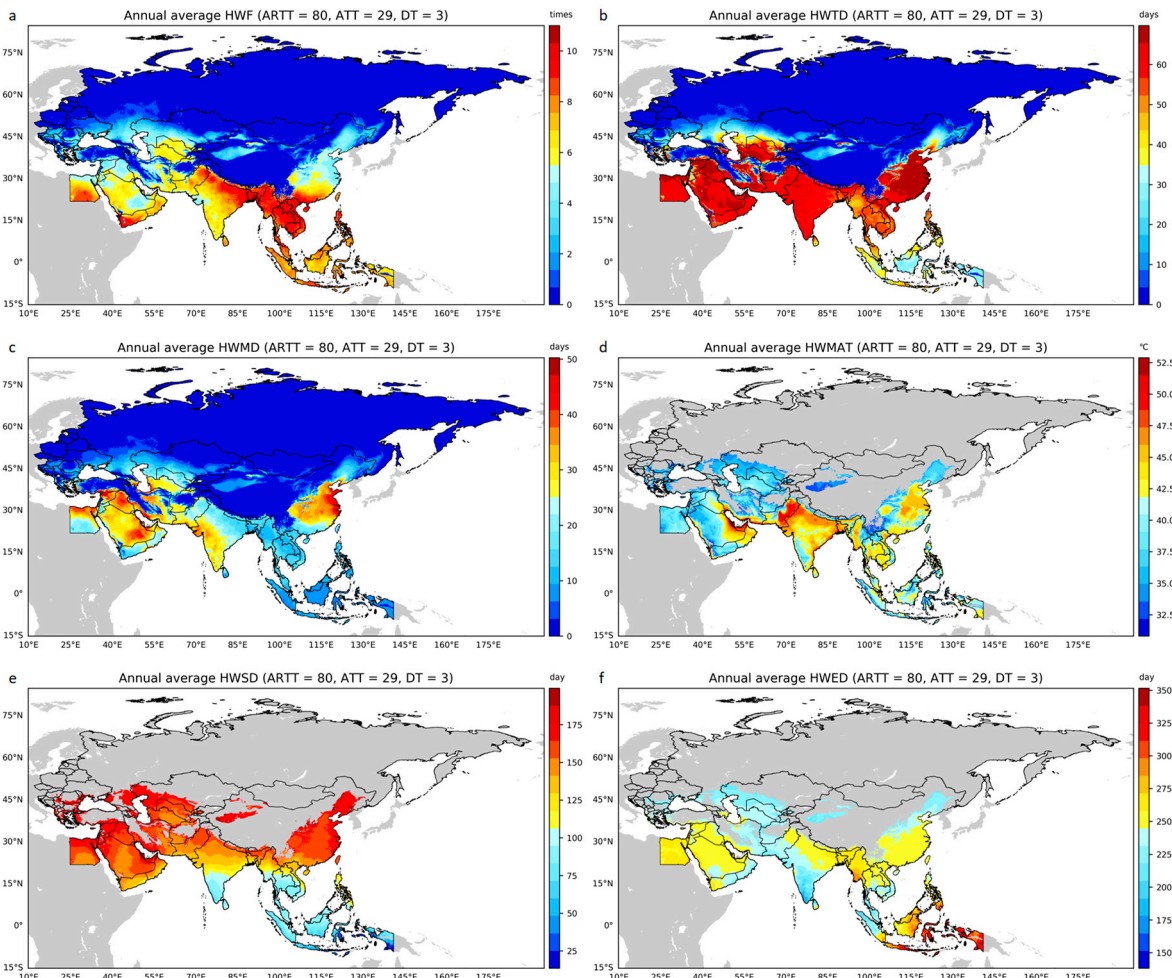

**Figure 7.** Annual average value of: (**a**) HWF; (**b**) HWTD; (**c**) HWMD; (**d**) HWMAT; (**e**) HWSD; (**f**) HWED.

To study the temporal variation of heat waves in the OBOR region over 30 years, we performed a linear regression of heat wave attributes (HWF, HWTD, and the other six attributes) on time. Figure 8 shows the slope of HWF, HWTD and the other six heat wave attributes in the regions passing the significance test (P value < 0.05) from 1989 to 2018. The values of HWF in 69.4% of the area, HWTD in 89.3% of the area and HWMD in 93.8% of the area are greater than 0, which means that most regions have more frequent and longer heat waves (Figure 8a,b). The frequency and duration of heat waves along the Black Sea coast are increasing rapidly (Figure 8a,b). HWMAT is greater than 0 in 79.1% of the area, which indicates that the heat wave intensity in most areas has an increasing trend. HWSD is less than 0 in 61.3% of the area and HWED is greater than 0 in 83.9% of the area, and highly overlaps in East China, North Southeast Asia and India, which indicates that the start date of the first heat wave is earlier and the end date of the last heat wave is later. This indicates that the heat wave duration is longer, which is consistent with the conclusions of Figure 8b,c. Overall, heat waves in the OBOR region are more frequent, more durable and more intense.

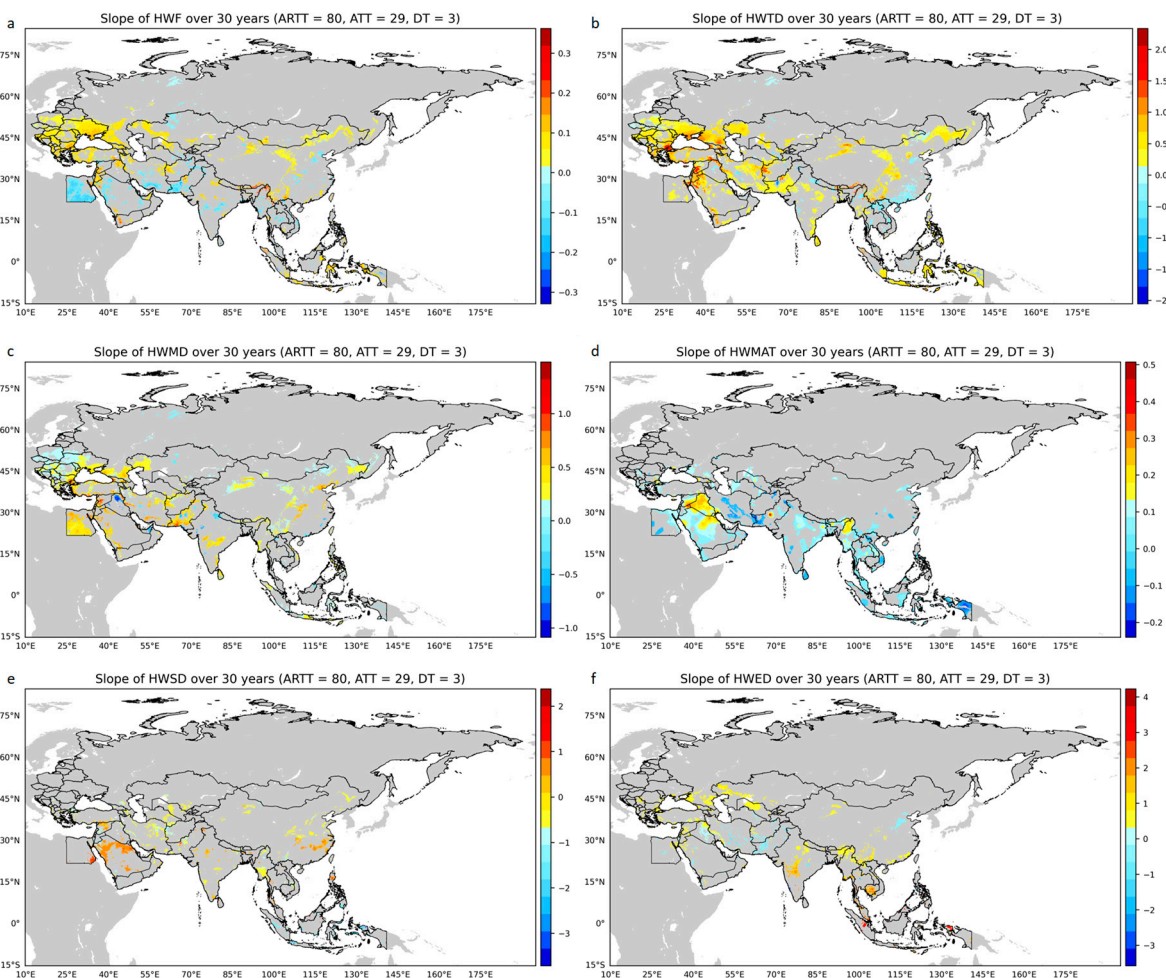

**Figure 8.** Slope over 30 years for: (**a**) HWF; (**b**) HWTD; (**c**) HWMD; (**d**) HWMAT; (**e**) HWSD; (**f**) HWED.

### 3.4. Heat Wave Risk Assessment

According to the risk triangle evaluation theory [53], there are three determinants of heat wave risk. For each determinant, we used different data to calculate the contribution of this determinant to heat wave risk (Table 2). We first standardize each data to 0-1, then use the same weight to calculate each determinant, and then standardize each determinant to calculate the heat wave risk with the same weight, the relationships of which are convenient to schematize in a mathematical form [53,80–82], defined as Equation (11). It should be noted that hazard is calculated using the 30-year average of HWF, HWTD and HWMAT, which is based on ARTT = 80, ATT = 29 and DT = 3. "Inverse (NDVI)" and "Inverse (GDP)" are due to the negative effect of NDVI and GDP on vulnerability.

$$\text{Hazard} = \left(\overline{\text{HWF}} + \overline{\text{HWTD}} + \overline{\text{HWMAT}}\right)/3 \tag{8}$$

$$\text{Exposure} = (\text{Night time lights} + \text{People under 14} + \text{People over 65})/3 \tag{9}$$

$$\text{Vulnerability} = (\text{Distance from water} + \text{Distance from hospital} + \text{Inverse}(\text{NDVI}) + \text{Inverse}(\text{GDP}))/3 \tag{10}$$

$$\text{Risk} = (\text{Hazard} + \text{Exposure} + \text{Vulnerability})/3 \tag{11}$$

Figure 9 shows the spatial distribution of hazard, exposure, vulnerability and risk in the OBOR region. The east of China, the north of Southeast Asia, the South Asia and the Persian Gulf are high- hazard areas, especially the north of India and the India-Pakistan border area (Figure 9a).



High-exposure areas are densely populated and economically developed urban areas, which are distributed in point or radial form, such as Beijing, Shanghai, Hong Kong, Moscow, New Delhi, etc., (Figure 9b). The areas with high vulnerability are those with limited water access, poor medical conditions, exposed surface and lagging economic development, such as Northwest China and far East and polar regions of Russia (Figure 9c). Combined with hazard, exposure and vulnerability, the high-risk areas are East China and North South Asia, and the risk exposure to cities located in these areas is higher (Figure 9d).

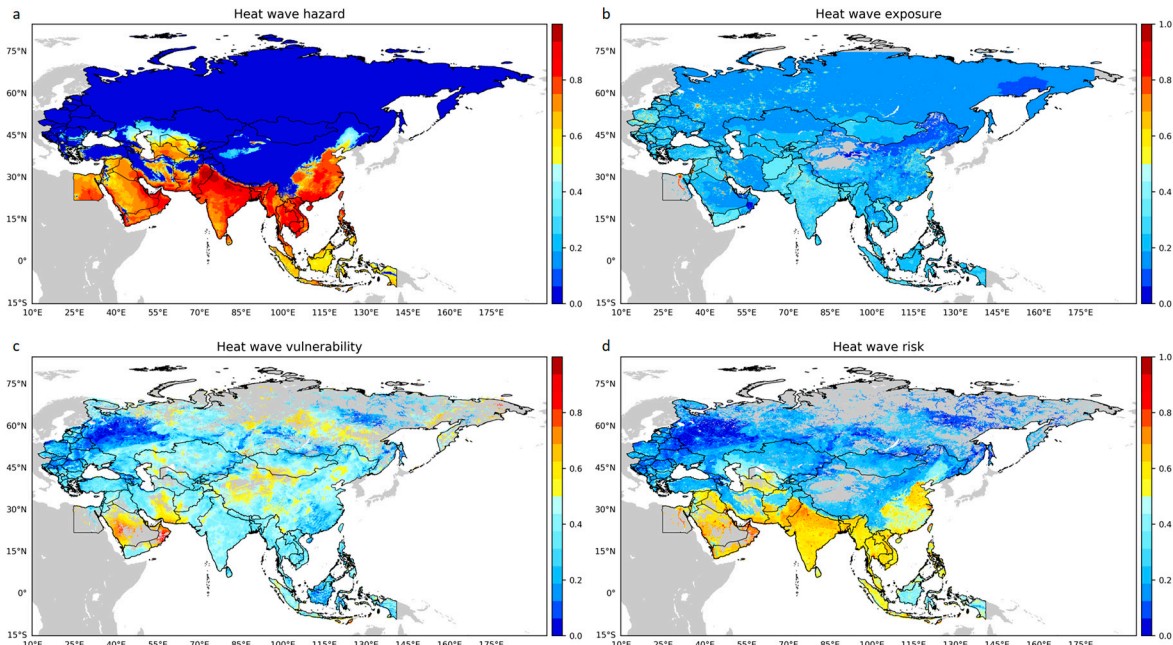

**Figure 9.** Heat wave risk assessment: (**a**) Heat wave hazard; (**b**) Heat wave exposure; (**c**) Heat wave vulnerability; (**d**) Heat wave risk.

## 4. Discussion

### 4.1. Comparison with Other Results

Our study found that the apparent temperature of the Tibetan Plateau and the Arctic region has an obvious upward trend, which is more obvious in autumn and winter. This trend has been widely reported [78,79]. Figure 6 shows that the average annual increase of apparent temperature in the Tibetan Plateau and the Arctic region has reached about 0.1 °C in the past 30 years. The Tibetan Plateau and the Arctic region are sensitive and vulnerable to climate change. They affect the global atmospheric circulation through dynamic forcing and thermal effects, and they are the driving force and amplifier of global climate change [83]. In recent decades, the sea ice area in the Arctic has been significantly reduced, reaching its peak in autumn, and large-scale warming has occurred in most parts of the Arctic, resulting in the loss of permafrost and the reduction of snow cover [78].

We further verify the performance of the heat wave dataset by comparing it with other results. In July 2010, western Russia experienced its hottest summer since records began in 1880. The heat wave caused an increase in deaths, drought, air pollution and reduced crop production, killing about 56,000 people and triggering about 500 wildfires [76,84]. It is reported that temperatures in parts of Russia soared to 42 °C, and fire and drought-inducing heat was expected to continue until at least August 12 (https://earthobservatory.nasa.gov/images/45069/heatwave-in-russia).

Based on the CHWT method (ARTT = 80, ATT = 29 and DT = 3), Figure 10 shows the six characteristics of the 2010 Russian heat wave. In 2010, a total of 3–5 heat waves were detected in western Russia (Figure 10a). The heat waves lasted for 25–40 days in total, and the longest one lasted for 20–30 days. Interestingly, 40 heat wave days were detected near Moscow, with the longest heat wave lasting for 30 days, the highest value in western Russia (Figure 10b,c). The extreme apparent temperature in western Russia is 34–38 °C (Figure 10g), the first heat wave occurs in the 170–200 days and the last heat wave ends in the 220–240 days. The HWSD and HWED are obviously layered (Figure 10h,i). Raei [29] first established the PDF (Probability Distribution Function) of daily temperature in different time windows, and used the temperature corresponding to different percentile thresholds to define the heat wave. Based on the time window of 21 days, the percentile of 90 and the duration of 3 days, Figure 10 shows the six characteristics of Russian heat wave in the same period. Compared with the results of the present study, the values of HWF and HWTD are relatively close, but the spatial distribution is quite different, and the high value areas are dotted and discrete (Figure 10d,e). The longest heat wave, 10–12 days, was detected near Moscow (Figure 10f). It is difficult to explain the spatial distribution of HWMAT, HWSD and HWED: firstly, it is unreasonable that the temperature is lower than 0 °C in the area with heat waves; secondly, it is not credible to detect a large area of heat wave in western Russia after the 300th day (Figure 10j,k,l). In general, the results of the present study are more consistent with the existing reports/studies, with better interpretability and resolution [76,84].

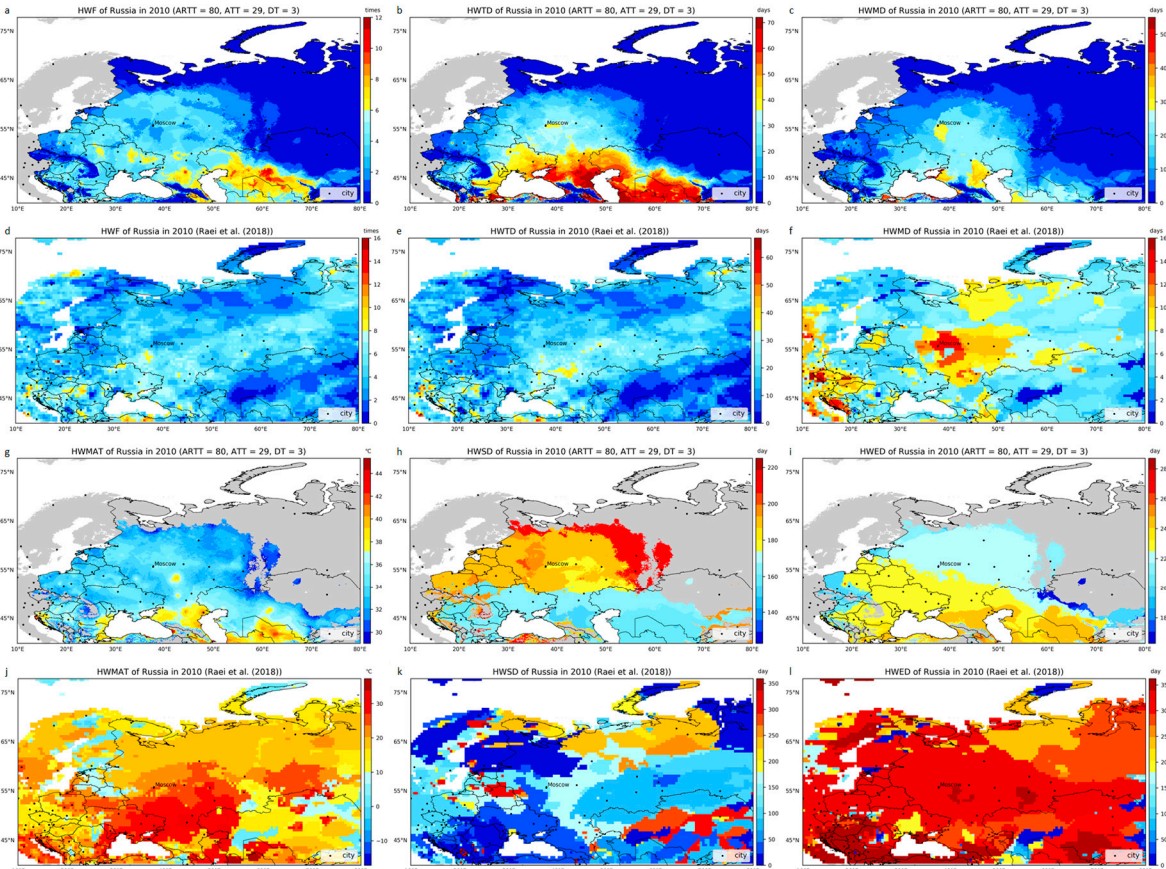

**Figure 10.** Comparison with other results: (**a–c,g–i**): HWF, HWTD, HWMD, HWMAT, HWSD, HWED under ARTT = 80, ATT = 29 and DT = 3; (**d–f,j–l**): HWF, HWTD, HWMD, HWMAT, HWSD, HWED (Raei et al. (2018)).

Li et al. mapped the global heat wave risk based on mortality and noted that high heat wave mortality risk areas are relatively scattered, distributed mainly in southern Asia, Europe, and the eastern part of North America at the grid level [31]. Figure 11 shows the heat wave risk mapped by this study. Through comparison with the evaluation results of the present study (Figure 9d), it is found that the two evaluation results show good consistency in East China, Southeast Asia and South Asia, and accurately identify high-risk areas such as North China, North India and Pakistan. The risk assessment results of Eastern Europe and Russia are quite different, mainly due to Li not considering the reduction effects of hospital and vegetation coverage on heat wave risk, while the heat wave hazard in this region is low (Figure 9a), the medical conditions are good (Figure 9b), and the vegetation coverage is high, so heat wave risk in this region is at a low level. By comparison, the risk assessment results of the present study are consistent with the existing studies, the assessment indicators are more comprehensive and the accuracy of the assessment results is higher, which can better reflect the heat wave risk level in the OBOR region.

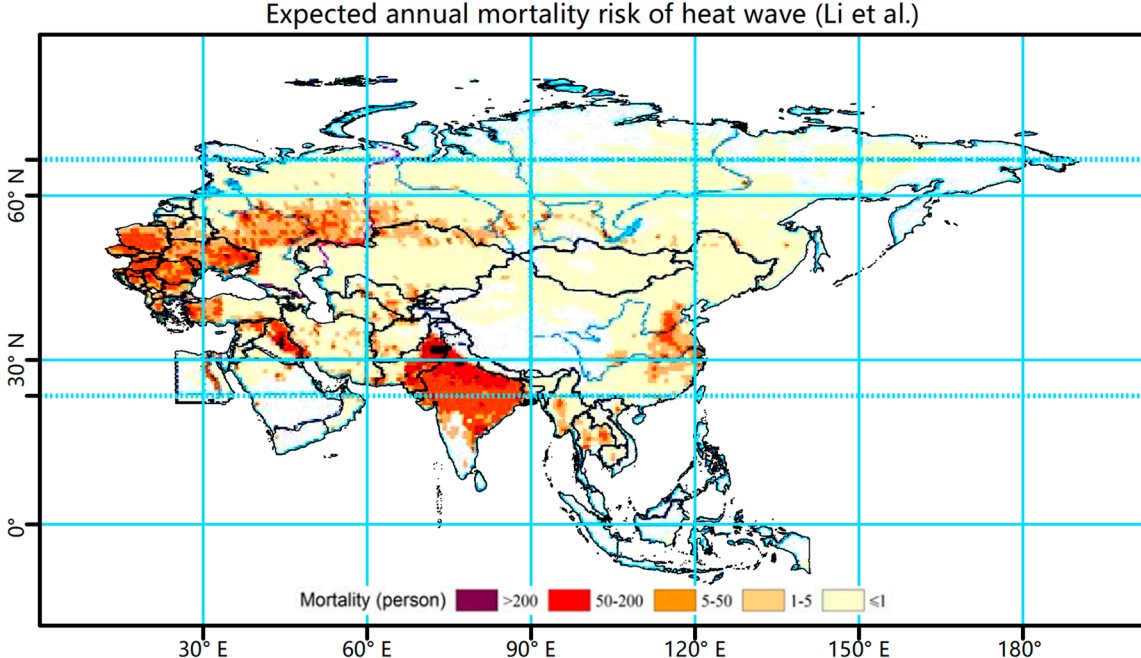

**Figure 11.** Heat wave risk assessment (Li et al.).

*4.2. Improvements and Limitations*

The main improvements of the present study are: first, unlike previous studies, which used air temperature to define heat waves, this study innovatively uses apparent temperature to define heat waves, which can better reflect the actual feelings of people in the environment; secondly, the CHWT method used in this study is based on different combinations of RTT and ATT, which makes the temperature threshold localized spatially and temporally to reflect the different adaptability of people in different regions to high temperature; finally, the present study uses multi-source data to assess heat wave risk. It comprehensively evaluates heat wave risk from three aspects: hazard, exposure and vulnerability. It not only considers the risk source, but also the bearing body and the risk reduction factors, so it can better reflect the risk level of different regions.

The main limitation is the importance of heat wave risk assessment factors (night light, population, water/hospital distribution, NDVI and GDP) was not differentiated. In fact, the contribution of each factor to heat wave risk is different, and the determinants of heat wave risk in different regions are also different. Therefore, when conducting a heat wave risk assessment, different factors should be used with different weights in different regions, which requires further research.

## 5. Conclusions

This study calculates the heat wave dataset of the OBOR region from 1989 to 2018 based on apparent temperature, and uses the heat wave, night lights, population, water bodies, hospitals, NDVI and GDP data to comprehensively evaluate the heat wave risk of the OBOR region. The spatiotemporal distribution and risk analysis show that the apparent temperature in most regions has a significant increase trend, and this trend is more obvious in the Tibetan Plateau and the Arctic region in autumn and winter; the frequency, duration and intensity of heat waves in eastern China, Southeast Asia and South Asia are at a relatively high level, with an average annual frequency of more than 6 times, lasting more than 60 days, and extreme apparent temperature reaching above 40 °C, and most regions have a rising trend; eastern China, northern South Asia, and some cities are high heat wave risk areas. In the process of continuing the development of the OBOR countries, the risk of heat waves must be fully considered. In areas with high heat wave risk, the government should guarantee the supply of water, electricity and medical facilities. Enterprises should invest cautiously. Passengers should also travel cautiously, and residents should pay more attention to the prevention of related diseases.

**Author Contributions:** Conceptualization, C.Y. and F.Y.; Data curation, C.Y., F.Y. and Y.Y.; Formal analysis, C.Y. and F.Y.; Funding acquisition, F.Y. and J.W.; Investigation, C.Y. and Y.Y.; Methodology, C.Y. and F.Y.; Project administration, F.Y. and J.W.; Resources, C.Y. and Y.Y.; Software, C.Y.; Supervision, F.Y. and J.W.; Validation, C.Y.; Visualization, C.Y.; Writing—original draft, C.Y.; Writing—review & editing, F.Y. and J.W. All authors have read and agreed to the published version of the manuscript.

**Funding:** This research was funded by the Strategic Priority Research Program of the Chinese Academy of Sciences, grant number XDA20030302, and Construction Project of China Knowledge Center for Engineering Sciences and Technology, grant number CKCEST-2019-3-6.

**Acknowledgments:** The present study used meteorological monitoring data and night light data from NCEI, elevation data from CIAT, population data from SEDAC, water body data from FAO, hospital data from OSM, NDVI data from NOAA, GDP data from DRYAD and heat wave data from Raei et al.

**Conflicts of Interest:** The authors declare no conflict of interest.

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
