# Peer review of "Spatiotemporal Distribution and Risk Assessment of Heat Waves Based on Apparent Temperature in the One Belt and One Road Region"

_remotesensing, doi:10.3390/rs12071174_

Round 1

Reviewer 1 Report

I have reviewed an earlier version of this manuscript, and I must certainly agree that you have done a great job in improving the manuscript (MS). It examines the spatiotemporal distribution of heat waves based on apparent temperature in the One Belt and One Road (Silk Route) region and assesses the associated risk to people. Daily meteorological data 1989-2018 is used to calculate the daily apparent temperature dataset, annual heat waves and their spatiotemporal distribution, including neighboring regions. A 0.1°× 0.1° resolution is used and the daily apparent temperature is based on ambient temperature, relative humidity (RH) and wind strength. The changes in frequency and duration of heat waves were analyzed for the regions studied. It is concluded that that the heat wave risk in most areas along the One Belt and One Road route is relatively high and that the heat wave risk should be considered when considering processes for further developing the One Belt and One Road region.

Title

The title is representative for the study.

Abstract

The Abstract describes what is done, is fairly well written and it presents the main conclusions of the study. I think that the trend of increasing temperatures, in particular in high mountain plateaus and in the arctic areas, ref line 1417 of the Conclusions section, could be highlighted some more? (This is in line with many other studies, some of which could be referred in the Discussion section.)

General comments

I think that the English language still needs some improvements, preferably by an English speaking person or a company offering light edits, etc. As a non-English speaking person, I am happy with the job Linda March have done for me at the www.goodenglishcompany.com. I also know that MDPI offers such services. Some examples of wordings needing improvement:

Line 36: …, 7 countries locate in OBOR

Line 39: … causing heavy casualties and

Line 98: 56,000 (while other places you write 56000)

Line 123: 2099?

Line 606: Heat wave hazard is a measure of the source of heat wave risk. (Is it a measure of the source? Please explain, or rewrite.)

Line 612: … which can decrease the risk of heat wave [51-53]. The "risk of heat wave" and the "heat wave risk" are indeed different expressions. Be careful not to mix "probability" and "risk", where the term risk contains both the probability of a heat wave and the outcome severity of the heat wave. Check this throughout, or state the way you understand the term risk.

Line 613 and some other lines: Error! Reference source not found. It looks like something is wrong with the pdf?

Line 659: The main influence object of heat wave disaster is human. Rewrite.

Line 661: … wind speed and humidity [58]. Best to write "relative humidity" in all similar expressions.

Line 680, Equation (5): Is the E used in the equation above?

Line 727: …, and high temperature processes for … processes?

Line 747: Accurate calculation results of apparent temperature are the premise… Rather: Accurate apparent temperatures are the premise…?

Line 749-750: should be: of March

Line 910: This should be Figure 2? (Correct that, and check the number of all succeeding figures.) In this figure, some points clearly deviate from the straight line. This is nearly always the case in real life investigations, but may I ask you to check if there could be an obvious reason for this deviation? A river close to the measurement station, narrow valley shielding the wind and making the area hotter than the neighborhood, or the like? (The reason I ask is that sometimes such deviations may contain information that can be used in the next generation models where such factors, if known, could maybe be better corrected for.)

Line 912-916: This is a very long sentence. Divide it in at least two parts.

Line 949-954: Very long sentence!

Line 970: Can you specify in the MS what the high value area means, e.g. the high heat wave risk area?

Line 979: The figure does not temporarily show something. It shows it permanently.

Line 982: Be considerate with the number of digits. 90.5%, etc would be better.

Line 1002: Reached what more than 9 times?

Line 1065: Be considerate with the number of digits so it reflects the actual precision.

Line 1066: … where there is little heat wave. Low heat wave frequency? Low heat wave severity? Low heat wave risk?

Line 1132: In each determinant, … For each determinant, …

Line 1237: … with difficult water use, … Rather "limited water access", or something like that?

Line 1240: … the risk of cities … Rather "the risk exposure to cities"…?

Figure numbers: e.g. Figure 8e1 and f1, is that allowed according to the Remote sensing Template?

Line 1393: In the Figure 9 caption, it is not according to the Template. Nor can I find a), b), c) and d) figures, only one. The text above the figure should rather be placed below the figure as part of the figure caption.

Line 1401: … humidity of apparent temperature …?

Line 1415: … in OBOR region, heat wave risk… Rather: …. in the OBOR region. The heat wave risk…

References

The Reference list has been greatly improved.

Overall evaluation

Your study is indeed an interesting read. The MS has been much improved. In my opinion, it now only needs a minor revision, including a language upgrade. I hope to receive an updated MS in the near future.

Author Response

Response to Reviewer 1 Comments

Thank you very much for your suggestions. The following reference (e.g. Line 126) are based the version that all revisions are accepted.

Point 1: Abstract:The Abstract describes what is done, is fairly well written and it presents the main conclusions of the study. I think that the trend of increasing temperatures, in particular in high mountain plateaus and in the arctic areas, ref line 1417 of the Conclusions section, could be highlighted some more? (This is in line with many other studies, some of which could be referred in the Discussion section.)

Response 1: Thank you for your confirmation. We did find that the apparent temperature of the Qinghai-Tibet Plateau and the polar regions tended to increase significantly in the autumn and winter seasons. According to your suggestions, we have added the corresponding part in the first paragraph of the discussion section (the first paragraph of section 4.2).

Point 2: General comments: I think that the English language still needs some improvements, preferably by an English speaking person or a company offering light edits, etc. As a non-English speaking person, I am happy with the job Linda March have done for me at the www.goodenglishcompany.com. I also know that MDPI offers such services.

Response 2: Thank you for your suggestion. Before the submission, we had commissioned AJE to perform an English editing, but still exposed some problems. My co-authors and I have tried our best to make changes based on your suggestions. If there is still a need for changes, we will consider entrusting MDPI to make more professional amendments.

Point 3: Line 36: …, 7 countries locate in OBOR …

Response 3: We rephrased this sentence into “From 1995 to 2015, among the 10 most severely meteorological disasters-affected countries, 7 are in the OBOR region”. (Line 40)

Point 4: Line 39: … causing heavy casualties and …

Response 4: We rephrased this sentence into “causing serious casualties and property losses”. (Line 43)

Point 5: Line 98: 56,000 (while other places you write 56000)

Response 5: We unified this number to "56,000". (Line 45 and 398)

Point 6: Line 123: 2099?

Response 6: This reference study is indeed the case, the author conducted a predictive study, and to avoid reader confusion, we rephrased this sentence into “Defrance conducted prediction studies and produced a data set of global extreme climatic indices annually and globally with a resolution of 0.5° × 0.5° from 1951 to 2099”. (Line 72)

Point 7: Line 606: Heat wave hazard is a measure of the source of heat wave risk. (Is it a measure of the source? Please explain, or rewrite.)

Response 7: We rephrased this sentence into “Heat wave hazard is a measure of the severity of heat wave events”. (Line 127)

Point 8: Line 612: … which can decrease the risk of heat wave [51-53]. The "risk of heat wave" and the "heat wave risk" are indeed different expressions. Be careful not to mix "probability" and "risk", where the term risk contains both the probability of a heat wave and the outcome severity of the heat wave. Check this throughout, or state the way you understand the term risk.

Response 8: We unified this phrase to "heat wave risk". (Line 134)

Point 9: Line 613 and some other lines: Error! Reference source not found. It looks like something is wrong with the pdf?

Response 9: The error has been revised. (Line 134)

Point 10: Line 659: The main influence object of heat wave disaster is human. Rewrite.

Response 10: We rephrased this sentence into “Heat waves directly cause discomfort and affect human health. The impact of heat waves on human health has been widely studied and reported”. (Line 186)

Point 11: Line 661: … wind speed and humidity [58]. Best to write "relative humidity" in all similar expressions.

Response 11: We unified this phrase to "relative humidity". (Line 189)

Point 12: Line 680, Equation (5): Is the E used in the equation above?

Response 12: The letter should be “P”. (Line 208)

Point 13: Line 727: …, and high temperature processes for … processes?

Response 13: We rephrased this sentence into “and a high temperature weather process that lasts for more than three days is called a heat wave”. (Line 213)

Point 14: Line 747: Accurate calculation results of apparent temperature are the premise… Rather: Accurate apparent temperatures are the premise…?

Response 14: We rephrased this sentence into “Accurate apparent temperatures are the premise of heat wave calculation”. (Line 238)

Point 15: Line 749-750: should be: of March

Response 15: It has been revised. (Line 238)

Point 16: Line 910: This should be Figure 2? (Correct that, and check the number of all succeeding figures.) In this figure, some points clearly deviate from the straight line. This is nearly always the case in real life investigations, but may I ask you to check if there could be an obvious reason for this deviation? A river close to the measurement station, narrow valley shielding the wind and making the area hotter than the neighborhood, or the like? (The reason I ask is that sometimes such deviations may contain information that can be used in the next generation models where such factors, if known, could maybe be better corrected for.)

Response 16: We revised the order of the pictures and explained how some points deviate from the line. (Line 256 and 265)

Point 17: Line 912-916: This is a very long sentence. Divide it in at least two parts.

Response 17: We rephrased this sentence into “Figure 2 and Table 3 show six cross validation results, the slope is higher than 0.99 and lower than 1.01, the R2 is higher than 0.96, and the result passed the significance test (P value < 0.01)”. (Line 262)

Point 18: Line 949-954: Very long sentence!

Response 18: We rephrased this sentence into “In this calculation result, the high value area of HWF is North India and North Southeast Asia (Figure 3b), the high value area of HWTD is southeast China , North Southeast Asia, South Asia, South Central Asia and Arabian Peninsula (Figure 3d), and the high value areas of HWMAT are eastern China and northern India (Figure 3f), which are consistent with other studies. Therefore, the calculation results based on ARTT can better reflect the actual spatial distribution of heat wave”. (Line 282)

Point 19: Line 970: Can you specify in the MS what the high value area means, e.g. the high heat wave risk area?

Response 19: We specified the high value as “top 20% of the value series”. (Line 280)

Point 20: Line 979: The figure does not temporarily show something. It shows it permanently.

Response 20: We rephrased the word “temporarily” to "timely". (Line 307)

Point 21: Line 982: Be considerate with the number of digits. 90.5%, etc would be better.

Response 21: We have kept one decimal place for these numbers. (Line 314)

Point 22: Line 1002: Reached what more than 9 times?

Response 22: We rephrased this sentence into “Among them, the frequency of heat waves in north of Southeast Asia and the northeast of India reached more than 9 times”. (Line 332)

Point 23: Line 1065: Be considerate with the number of digits so it reflects the actual precision.

Response 23: We have kept one decimal place for these numbers. (Line 351)

Point 24: Line 1066: … where there is little heat wave. Low heat wave frequency? Low heat wave severity? Low heat wave risk?

Response 24: This sentence is deleted.

Point 25: Line 1132: In each determinant, … For each determinant, …

Response 25: It has been revised. (Line 364)

Point 26: Line 1237: … with difficult water use, … Rather "limited water access", or something like that?

Response 26: We rephrased this sentence into “The areas with high vulnerability are those with limited water access”. (Line 376)

Point 27: Line 1240: … the risk of cities … Rather "the risk exposure to cities"…?

Response 27: We rephrased this sentence into “and the risk exposure to cities located in these areas is higher”. (Line 379)

Point 28: Figure numbers: e.g. Figure 8e1 and f1, is that allowed according to the Remote sensing Template?

Response 28: We renumbered the pictures using letters. (Line 418)

Point 29: Line 1393: In the Figure 9 caption, it is not according to the Template. Nor can I find a), b), c) and d) figures, only one. The text above the figure should rather be placed below the figure as part of the figure caption.

Response 29: We renamed the picture " Heat wave risk assessment (Li et al.)". (Line 437)

Point 30: Line 1401: … humidity of apparent temperature …?

Response 30: We rephrased this sentence into “this study innovatively uses apparent temperature to define heat waves”. (Line 440)

Point 31: … in OBOR region, heat wave risk… Rather: …. in the OBOR region. The heat wave risk…

Response 31: We rephrased this sentence into “we studied the spatiotemporal distribution of heat wave in the OBOR region, the heat wave risk in this area is also evaluated”. (Line 455)

Reviewer 2 Report

Review: Spatiotemporal distribution and risk assessment of heat waves based on apparent temperature in the One Belt and One Road Region

In their work, the authors develop a set of site-specific thresholds for heat waves based on a weather station network interpolated into a regular grid. The threshold uses apparent temperature, which considers not just air temperature, but humidity and wind speed to estimate the “felt” temperature by people. The work is significant because it addresses the need for spatially and temporally comprehensive heat wave data across the OBOR region, where large investments and development are being directed, and risk to these hazards is bound to increase.

I found the article suitable for publication, given the author performs major changes to their manuscript. Although the organization of the work was easy to follow, I found gaps in the explanation of the methods that seriously hinder reproducibility and transparency. I also found significant gaps in their introduction to heat wave science, which basically ignores the physical processes that lead to heat waves and their interactions with land cover. This omission is particularly important because the OBOR region is being actively developed, presumably leading to fast land use change.

I also found several minor mistakes or errors that should be addressed. Some are due to document formatting issues (e.g., missing references) while others are related to unclear language or writing. These issues should be addressed to avoid misinterpretations of the work. I detail more specific comments below:

Major Comments

The method used to compute RTT needs more detail. In the manuscript, the authors state that:

(Line 734-744) : “… we establish a PDF (Probability Distribution Function) of the historical temperature of a certain day in 1989-2018, and select the temperature corresponding to different percentiles as the RTT to judge heat waves, which is defined as Climatological Relative Temperature Threshold (CRTT); then, when the temperature of a certain day is a higher value in the temperature series of this year, it also reflects the possibility of extreme high temperature.”

What is the percentile threshold used here? The process described here essentially states that the authors “choose a percentile value and then compute heat wave days according to apparent temperatures that exceed the percentile of this threshold at each grid point”.

(Line 749) When validating their model results, the authors pick a very small sample size of 4 single days. When compared to their multi-decadal time study period, this seems woefully inadequate. What’s more, their validation set is unlikely to find the variety of temperature ranges across the warm season.

Lines 979-990: Are these trends statistically significant everywhere on the OBOR region? Please conduct the appropriate significance test.

Section 3.4: In their framing of vulnerability, the authors focus on Distance from water, distance from hospital, and the inverse of the NDVI as the main factors. Why are these the only ones? Did the authors consider other indicators? In other countries, studies have shown that income levels, and access to cooling equipment might also play an important role in heat-related mortality (Rosenthal et al 2014, Madrigano et al 2015, Ito et al 2018). Although these particular indicators may not be as relevant in the OBOR region, there might be other important socioeconomic metrics that should be considered.

Madrigano J, Ito K, Johnson S, Kinney P L and Matte T 2015 A Case-Only Study of Vulnerability to Heat Wave–RelatedMortality in New York City (2000–2011) Environmental Health Perspectives 123 672–8

Ito K, Lane K and Olson C 2018 Equitable Access to Air Conditioning: A City Health Department’s Perspective on Preventing Heat-related Deaths Epidemiology 29 749–52

Rosenthal J, Kinney P L and Metzger K B 2014 Intra-urban vulnerability to heat-related mortality in New York City, 1997–2006 Health & Place 30 45–60

Guo Y, Gasparrini A, Armstrong B G, Tawatsupa B, Tobias A, Lavigne E, Coelho M de S Z S, Pan X, Kim H, Hashizume M, Honda Y, Guo Y-L L, Wu C-F, Zanobetti A, Schwartz J D, Bell M L, Scortichini M, Michelozzi P, Punnasiri K, Li S, Tian L, Garcia S D O, Seposo X, Overcenco A, Zeka A, Goodman P, Dang T N, Dung D V, Mayvaneh F, Saldiva P H N, Williams G and Tong S 2017 Heat Wave and Mortality: A Multicountry, Multicommunity Study Environmental Health Perspectives 125 Online: http://ehp.niehs.nih.gov/EHP1026

Lines 1409-1412: This method also assumes that risk changes linearly with the selected factors, which might not be the case.

Minor Comments

In the Abstract, line 21, the sentence “The heat waves in China, South Asia and Southeast Asia have high frequency, long duration and high intensity...” is very general and does not provide much information to the reader. Are they higher than in other regions? Consider qualifying this claim (as you do in the introduction).

Line 112-115: Although these are all active areas of research, the claim is misleading because there are of course other significant research areas. Some include work on synoptic processes that lead to heat waves. Horton et al (Horton et al 2016) has a nice review of some of this work, although surely much has been published yet. With regards to relationships between development and heat waves, there is a growing but significant body of work addressing interactions between local and large scale signals related to heat waves. Please look at the following work:

Horton R M, Mankin J S, Lesk C, Coffel E and Raymond C 2016 A Review of Recent Advances in Research on Extreme Heat Events Current Climate Change Reports 2 242–59

An N, Dou J, González J E, Bornstein R D, Miao S and Li L 2020 An observational case study of synergies between an intense heatwave and the urban heat island in Beijing J. Appl. Meteor. Climatol. JAMC-D-19-0125.1

Ao X, Wang L, Zhi X, Gu W, Yang H and Li D 2019 Observed Synergies between Urban Heat Islands and Heat Waves and Their Controlling Factors in Shanghai, China J. Appl. Meteor. Climatol. 58 1955–72

Founda D and Santamouris M 2017 Synergies between Urban Heat Island and Heat Waves in Athens (Greece), during an extremely hot summer (2012) Scientific Reports 7 Online: http://www.nature.com/articles/s41598-017-11407-6

Khan H S, Paolini R, Santamouris M and Caccetta P 2020 Exploring the Synergies between Urban Overheating and Heatwaves (HWs) in Western Sydney Energies 13 470

Li D and Bou-Zeid E 2013 Synergistic interactions between urban heat islands and heat waves: The impact in cities is larger than the sum of its parts Journal of Applied Meteorology and Climatology 52 2051–64

Ortiz L E, Gonzalez J E, Wu W, Schoonen M, Tongue J and Bornstein R 2018 New York City Impacts on a Regional Heat Wave Journal of Applied Meteorology and Climatology 57 837–51

Li D, Sun T, Liu M, Yang L, Wang L and Gao Z 2015 Contrasting responses of urban and rural surface energy budgets to heat waves explain synergies between urban heat islands and heat waves Environmental Research Letters 10 054009

Line 123: This claim is a bit confusing. What do the authors mean when they say that existing data production methods only use a single source?

Line 386: “date set” should be “dataset”

Line 394: There is a missing reference here.

Line 608: The meaning of “carrier” is unclear here.

Line 613: Missing reference

Line 622: Why couldn’t kriging with regression also be used here?

Line 659: This list misses incoming solar radiation as an important parameter

Line 922: It is unclear what the authors mean by the sentence “To verify the availability of the…”

Line 1246: Please check the use of the word “availability”. Authors might be referring to model performance or skill.

Line 1387: Missing reference

Author Response

Response to Reviewer 2 Comments

Thank you very much for your suggestions. The following reference (e.g. Line 126) are based the version that all revisions are accepted.

Point 1: The method used to compute RTT needs more detail. In the manuscript, the authors state that:

(Line 734-744) : “… we establish a PDF (Probability Distribution Function) of the historical temperature of a certain day in 1989-2018, and select the temperature corresponding to different percentiles as the RTT to judge heat waves, which is defined as Climatological Relative Temperature Threshold (CRTT); then, when the temperature of a certain day is a higher value in the temperature series of this year, it also reflects the possibility of extreme high temperature.”

What is the percentile threshold used here? The process described here essentially states that the authors “choose a percentile value and then compute heat wave days according to apparent temperatures that exceed the percentile of this threshold at each grid point”.

Response 1: Thank you for your suggestion. Following your suggestion, we have simplified the description of the method to highlight its most important parts. The new description is “Therefore, for the selected date, we rank the apparent temperature of each grid point in 1989-2018, and then select the temperature corresponding to different percentiles as the RTT to judge heat waves”. (Line 220)

Point 2: (Line 749) When validating their model results, the authors pick a very small sample size of 4 single days. When compared to their multi-decadal time study period, this seems woefully inadequate. What’s more, their validation set is unlikely to find the variety of temperature ranges across the warm season.

Response 2: We improved the verification part, cross-validated the apparent temperature interpolation results of the entire time series, and showed the verification results for six days in the manuscript. (Line 238)

Point 3: Lines 979-990: Are these trends statistically significant everywhere on the OBOR region? Please conduct the appropriate significance test.

Response 3: We performed a significance test on the trend analysis section, and the image and analysis sections only targeted those areas that passed the significance test. (Line 307 and 347)

Point 4: Section 3.4: In their framing of vulnerability, the authors focus on Distance from water, distance from hospital, and the inverse of the NDVI as the main factors. Why are these the only ones? Did the authors consider other indicators? In other countries, studies have shown that income levels, and access to cooling equipment might also play an important role in heat-related mortality (Rosenthal et al 2014, Madrigano et al 2015, Ito et al 2018). Although these particular indicators may not be as relevant in the OBOR region, there might be other important socioeconomic metrics that should be considered.

Response 4: Your suggestion is very constructive, and vulnerability is highly related to economic level and cooling conditions (such as air conditioning). Therefore, we added economic factors (GDP) to the vulnerability assessment framework. However, due to the large scope of the study area, it is difficult to obtain the use of cooling equipment in the entire region. We expect to consider this factor in a small-scale and high-precision evaluation. (Line 362)

Point 5: Lines 1409-1412: This method also assumes that risk changes linearly with the selected factors, which might not be the case.

Response 5: Our expression caused misunderstanding. In fact, what we want to say is that each factor should have different importance in the risk evaluation, and the importance of the same factor in different regions is also different. We rephrased this sentence into “In fact, the contribution of each factor to heat wave risk is different, and the determinants of heat wave risk in different regions are also different”. (Line 449)

Point 6: In the Abstract, line 21, the sentence “The heat waves in China, South Asia and Southeast Asia have high frequency, long duration and high intensity...” is very general and does not provide much information to the reader. Are they higher than in other regions? Consider qualifying this claim (as you do in the introduction).

Response 6: We rephrased this sentence into “China, South Asia and Southeast Asia are suffering the most serious heat waves in the OBOR region, with an average of more than 6 heat waves, lasting for more than 60 days and the extreme apparent temperature has reached over 40℃. Additionally, the frequency, duration and intensity of heat waves have been confirmed to increase continuously”. (Line 21)

Point 7: Line 112-115: Although these are all active areas of research, the claim is misleading because there are of course other significant research areas. Some include work on synoptic processes that lead to heat waves. Horton et al (Horton et al 2016) has a nice review of some of this work, although surely much has been published yet. With regards to relationships between development and heat waves, there is a growing but significant body of work addressing interactions between local and large scale signals related to heat waves.

Response 7: We have added two aspects of related studies: (1) the mechanism that caused the heat wave ; (2) synergy between urban heat islands and heat waves. (Line 58)

Point 8: Line 123: This claim is a bit confusing. What do the authors mean when they say that existing data production methods only use a single source?

Response 8: We rephrased this sentence into “they have more restrictions on the data source (such as the format of the data)”. (Line 73)

Point 9: Line 386: “date set” should be “dataset”

Response 9: The error has been revised. (Line 110)

Point 10: Line 394: There is a missing reference here.

Response 10: The error has been revised. (Line 117)

Point 11: Line 608: The meaning of “carrier” is unclear here.

Response 11: We rephrased this sentence into “Heat wave exposure is the degree to which people, livelihoods, and economy etc. may be adversely affected.”. (Line 129)

Point 12: Line 613: Missing reference

Response 12: The error has been revised. (Line 135)

Point 13: Line 622: Why couldn’t kriging with regression also be used here?

Response 13: Due to the large area of the study area, the vertical zone of temperature must be considered, and it is difficult to reflect this change directly using traditional interpolation methods. For the regression Kriging, although it can get better interpolation results, it is very time consuming. The interpolation method we use has been proven to have a good interpolation effect, so the advantage of using regression kriging is less than the disadvantage due to time cost. We will still consider using higher performance equipment and algorithms to improve interpolation accuracy.

Point 14: Line 659: This list misses incoming solar radiation as an important parameter

Response 14: We rephrased this sentence into “The cold and hot feeling to the external environment of human body is affected by the comprehensive influence of air temperature, wind speed, relative humidity and solar radiation”. (Line 188)

Point 15: Line 922: It is unclear what the authors mean by the sentence “To verify the availability of the…”

Response 15: We rephrased this sentence into “To verify the performance of the heat wave data set”. (Line 274)

Point 16: Line 1246: Please check the use of the word “availability”. Authors might be referring to model performance or skill.

Response 16: We rephrased this sentence into “We further verify the performance of the heat wave data set by comparing it with other results.”. (Line 396)

Point 17: Line 1387: Missing reference

Response 17: The error has been revised. (Line 432)

Reviewer 3 Report

The manuscript presents an analysis of the risk assessment related to heatwave occurrences in the very extended area of the OBOR region. To this scope, the authors use daily meteorological data from 2,833 monitoring stations in the OBOR region to calculate annual heatwave for the period 1989-2018. They also build a spatial-temporal data set.

This is an important topic and I believe that the authors have done a useful analysis. However, there are some issues related to the paper that should be addressed:

  • rigor quality has to be improved;
  • methods require additional specifications.

Introduction

The authors present a well-structured framework of what is already known about the heatwave computation and the risk associated with them. However, the motivation of the heatwave dataset building is not clear. In particular, it is not clear to me if the authors intend to build the heatwave dataset to make an open-source (or restricted) release of it or if they want to illustrate how they compute heat waves to assess the risk related to them.

Materials and Methods

The study methods are valid and reliable; however, their presentation is not clear. I suggest the authors re-think the structure of the section and to re-write the technical parts. There are not enough details to replicate the study. In particular, they should introduce more mathematical notation relatively to the statistical approach of data imputation, data interpolation and trend assessment. For instance, at line 390, the authors introduce the missForest approach to process missing data, however the method is not discussed in details. Instead, the authors should give details on the values of parameters set and let the reader assess the efficiency of the imputation procedure. A similar suggestion I would give about the interpolation procedure: details should be given about the applied Kriging procedure, for example, which model of variogram has been applied, what are the estimates of sill, nugget, and range. In my opinion, these improvements would give merit to this paper.

Results

For the spatial interpolation, an assessment of the model performance is given in par. 3.1 by producing a measure of the deviation of predicted values of apparent temperature from real values. Here, the authors should explain how do they choose the 100-monitoring station used for validation: what is their spatial distribution? Are they located at different elevations? Since the spatial domain of the study is very large and the number of the stations very huge, it seems to me that 100 stations are not enough to account for predictive performance (I would say that the number of validation time series could be around 10-15% of the 2,833 stations). Furthermore, a rigorous measure of the uncertainty associated to the interpolation method should comprise the entire time series and not only the four days presented in Figure 1 and Table 2. The four time series of Figure 1 can remain for illustration purposes. Also, a uncertainty map of the entire spatial domain would be useful to visualize those areas where the interpolation method has a lower performance and, consequently, to a better evaluation of the heat wave risk at a spatial level.

To summarize, the content of the paper is suspended between the building of the dataset, the Spatio-temporal distribution of heat waves, and the evaluation of heatwave risk. I would rather divide the content of this very extended and demanding work into two papers: the one composed of the dataset building approach and its climatological analysis, the other dedicated to the assessment of heatwave risk.

Moreover, the understanding is further hampered by several language issues that need to be corrected. Finally, I strongly encourage the authors to revise the paper and have the merit of their big effort.

Specific major issues

  • The abstract should be re-written.
  • The imputation method should be discussed in detail by reporting both the value of parameters and the quantification of the procedure’s effectiveness.
  • Paragraph 2.2.3 is not clear. Its methodological part should be written in a more scientific manner. In fact, although the theoretical concepts are correct, it is quite hard to follow their development. For instance, in line 733 the authors present the methodology to calculate deviations from “historical temperature”, then they declare that the values of daily historical temperature come from a PDF. Here, I would rather present the entire procedure with some mathematical notation. Furthermore, the recurrence of a great number of acronyms for the heatwave indexes makes difficult to analyze the results.
  • Section 3.2 needs to be implemented with a rigorous trend analysis in order to present maps of the apparent temperature trend. Have the authors made a test of the significance of linear trend parameters?
  • For Section 3.3, the same argument of precedent item applies.

I have also some specific comments:

  • In the Introduction, the authors should be aware that the 1.5° threshold has been fixed as the limit in order to set a framework of plans the human activity should follow to keep global warming at the fixed level of increase rather than an increased level of temperature caused by human activity;
  • In the Introduction, a brief description of the regions belonging to the OBOR region would be useful to the reader;
  • In Table 2, the authors refer to “apparent temperature”, then they use Ta (air temperature °C) to classify its perception. Please, clarify the classification approach.
  • At lines 393 and 613, the reference is missing;
  • Revise reference 57 (is it a book?);
  • At line 751, the corresponding relative value of the 100 monitoring stations would let the reader know the percentage of the validation set respect to the training set;
  • At line 914, both RMSE and MAE would be more informative if their values were normalized with respect to the maximum (minimum).

Author Response

Response to Reviewer 3 Comments

Thank you very much for your suggestions. The following reference (e.g. Line 126) are based the version that all revisions are accepted.

Point 1: Introduction: The authors present a well-structured framework of what is already known about the heatwave computation and the risk associated with them. However, the motivation of the heatwave dataset building is not clear. In particular, it is not clear to me if the authors intend to build the heatwave dataset to make an open-source (or restricted) release of it or if they want to illustrate how they compute heat waves to assess the risk related to them.

Response 1: Thank you for your suggestion. Unlike most previous studies based on air temperature, this study uses apparent temperature to study heat waves, so corresponding data production work must be performed, which is also an important aspect of this study's innovation. The data will be published publicly. In the introduction, we made this clearer: " In order to assess the heat wave risk in this area on a more precise level, we constructed the heat wave data set and evaluated heat wave risk in the OBOR region using multi-source data…. The data set is now available on IKCEST (International Knowledge Center for Engineering Sciences and Technology)". (Line 86 and 90)

Point 2: Materials and Methods: The study methods are valid and reliable; however, their presentation is not clear. I suggest the authors re-think the structure of the section and to re-write the technical parts. There are not enough details to replicate the study. In particular, they should introduce more mathematical notation relatively to the statistical approach of data imputation, data interpolation and trend assessment. For instance, at line 390, the authors introduce the missForest approach to process missing data, however the method is not discussed in details. Instead, the authors should give details on the values of parameters set and let the reader assess the efficiency of the imputation procedure. A similar suggestion I would give about the interpolation procedure: details should be given about the applied Kriging procedure, for example, which model of variogram has been applied, what are the estimates of sill, nugget, and range. In my opinion, these improvements would give merit to this paper.

Response 2: In the second part, we introduced the data imputation process based on missforest and the setting of related parameters in the interpolation process in detail. (Line 140 and 177) The trend analysis section has also been improved through significance test.

Point 3: Results: For the spatial interpolation, an assessment of the model performance is given in par. 3.1 by producing a measure of the deviation of predicted values of apparent temperature from real values. Here, the authors should explain how do they choose the 100-monitoring station used for validation: what is their spatial distribution? Are they located at different elevations? Since the spatial domain of the study is very large and the number of the stations very huge, it seems to me that 100 stations are not enough to account for predictive performance (I would say that the number of validation time series could be around 10-15% of the 2,833 stations). Furthermore, a rigorous measure of the uncertainty associated to the interpolation method should comprise the entire time series and not only the four days presented in Figure 1 and Table 2. The four time series of Figure 1 can remain for illustration purposes. Also, a uncertainty map of the entire spatial domain would be useful to visualize those areas where the interpolation method has a lower performance and, consequently, to a better evaluation of the heat wave risk at a spatial level.

Response 3: We improved the verification part, 15% monitoring stations are used to cross-validate the apparent temperature interpolation results of the entire time series, and showed the verification results for six days in the manuscript. (Line 238)

Point 4: To summarize, the content of the paper is suspended between the building of the dataset, the Spatio-temporal distribution of heat waves, and the evaluation of heatwave risk. I would rather divide the content of this very extended and demanding work into two papers: the one composed of the dataset building approach and its climatological analysis, the other dedicated to the assessment of heatwave risk.

Response 4: Thank you for your suggestion. We will deepen the work in these two aspects in the follow-up research. At the same time, we hope that our improvements will coordinate the content in this manuscript.

Point 5: Moreover, the understanding is further hampered by several language issues that need to be corrected. Finally, I strongly encourage the authors to revise the paper and have the merit of their big effort.

Response 5: Thank you for your suggestion. Before the submission, we had commissioned AJE to perform an English editing, but still exposed some problems. My co-authors and I have tried our best to make changes based on your suggestions. If there is still a need for changes, we will consider entrusting MDPI to make more professional amendments.

Point 6: The abstract should be re-written.

Response 6: We rewritten the results section in the summary and quantified the results to provide more information. (Line 21)

Point 7: The imputation method should be discussed in detail by reporting both the value of parameters and the quantification of the procedure’s effectiveness.

Response 7: In Section 2.2.1, we described the data imputation process based on missforest in detail, and confirmed the performance of missforest in data imputation by referring to other studies. (Line 140)

Point 8: Paragraph 2.2.3 is not clear. Its methodological part should be written in a more scientific manner. In fact, although the theoretical concepts are correct, it is quite hard to follow their development. For instance, in line 733 the authors present the methodology to calculate deviations from “historical temperature”, then they declare that the values of daily historical temperature come from a PDF. Here, I would rather present the entire procedure with some mathematical notation. Furthermore, the recurrence of a great number of acronyms for the heatwave indexes makes difficult to analyze the results.

Response 8: Thank you for your suggestion. Following your suggestion, we have simplified the description of the method to highlight its most important parts. The new description is “Therefore, for the selected date, we rank the apparent temperature of each grid point in 1989-2018, and then select the temperature corresponding to different percentiles as the RTT to judge heat waves”. (Line 220) In addition, we provide an abbreviation lookup table to improve readability. (Line 103)

Point 9: Section 3.2 needs to be implemented with a rigorous trend analysis in order to present maps of the apparent temperature trend. Have the authors made a test of the significance of linear trend parameters? For Section 3.3, the same argument of precedent item applies.

Response 9: We performed a significance test on the trend analysis section, and the image and analysis sections only targeted those areas that passed the significance test. (Line 307 and 347)

Point 10: In the Introduction, the authors should be aware that the 1.5° threshold has been fixed as the limit in order to set a framework of plans the human activity should follow to keep global warming at the fixed level of increase rather than an increased level of temperature caused by human activity;

Response 10: With reference to the IPCC report, we have revised this sentence: In the Intergovernmental Panel on Climate Change (IPCC) special report, human activities are estimated to have caused approximately 1.0℃ of global warming above pre-industrial levels, with a likely range of 0.8℃ to 1.2℃. (Line 34)

Point 11: In the Introduction, a brief description of the regions belonging to the OBOR region would be useful to the reader;

Response 11: We have added a description of the OBOR region: The OBOR region involves 3 continents, more than 66 countries and regions and approximately 4.4 billion people, with frequent natural disasters, highly concentrated populations, and fragile ecological environments. (Line 38)

Point 12: In Table 2, the authors refer to “apparent temperature”, then they use Ta (air temperature °C) to classify its perception. Please, clarify the classification approach.

Response 12: We deleted this table because subsequent analysis does not need to classify apparent temperatures. In fact, the classification method is based on the golden section.

Point 13: At lines 393 and 613, the reference is missing;

Response 13: The error has been revised. (Line 117 and 135)

Point 14: Revise reference 57 (is it a book?);

Response 14: The reference has been updated. (Line 166)

Point 15: At line 751, the corresponding relative value of the 100 monitoring stations would let the reader know the percentage of the validation set respect to the training set;

Response 15: We used 15% of the available sites as a validation set for cross-validation. (Line 243)

Point 16: At line 914, both RMSE and MAE would be more informative if their values were normalized with respect to the maximum (minimum).

Response 16: Using Maximum-Minimum Normalization Method, we calculated the Normalized Mean Absolute Error (NMAE) and Normalized Root Mean Square Error (NRMSE). (Line 264)

Round 2

Reviewer 2 Report

The authors have edited the manuscript considerably, improving its clarity and the robustness of their analysis.

Although much improved, there are some points in my previous review that I found were not adequately addressed, listed below:

In Point 1, the authors did not fully respond to the question I posed regarding the selection of temperature percentiles for heatwaves (CRTT). Although there is mention of using CRTT of 90th percentile of temperature, is there a particular reason for doing so? There is also mention of using various percentiles, but the analysis only mentions a CRTT of 90.

Also, one limitation of using the Humidex, or similar apparent temperature measures that only use humidity and temperature, is that they do not account for some other factors that may determine the experience of heat. These include solar radiation and wind speed. The author included these in their review of the topic, but should also note it as a limitation of their apparent temperature metric as well.

Author Response

Response to Reviewer 2 Comments

Thank you very much for your suggestions. The following reference line (e.g. Line 126) are based the version that all revisions are accepted.

Point 1: In Point 1, the authors did not fully respond to the question I posed regarding the selection of temperature percentiles for heatwaves (CRTT). Although there is mention of using CRTT of 90th percentile of temperature, is there a particular reason for doing so? There is also mention of using various percentiles, but the analysis only mentions a CRTT of 90.

Response 1: Thank you for your suggestion. "Different percentiles" here is a description of the CHWT method, which aims to show that our method can use different percentile thresholds to adapt to different heat wave standards. We describe this more clearly: CHWT allows setting different percentile thresholds to accommodate different heat wave standards (Line 226). For the reasons for choosing the 90 threshold, we have explained and added reference studies: In this study, by referring to other work and comparing the results of different thresholds (CRTT = 85 and 90), we found that CRTT = 90 best reflects the actual state of heat waves (Line 227).

Point 2: Also, one limitation of using the Humidex, or similar apparent temperature measures that only use humidity and temperature, is that they do not account for some other factors that may determine the experience of heat. These include solar radiation and wind speed. The author included these in their review of the topic, but should also note it as a limitation of their apparent temperature metric as well.

Response 2: In this study, in order to avoid misleading readers, we have comprehensively listed the four factors that affect apparent temperature (air temperature, wind speed, relative humidity and solar radiation). However, based on the existing calculation methods and data accessibility of apparent temperature, the calculation of apparent temperature only considers the first three factors, but this is still better than the heat wave calculation method that only considers the air temperature. We point this out more clearly: Based on the existing calculation methods of apparent temperature and the accessibility of data, this study mainly considers the first three factors. As apparent temperature takes into account more environmental factors than air temperature, it can more accurately reflect the cold and hot degree of the external environment (Line 192).

Reviewer 3 Report

Dear Authors,

thank you for your responses, which I consider highly reasonable and acceptable. Also, the readability of the paper has been significantly improved. 

I have two minor points regarding either the interpolation method or the linear trend estimation:

1) KRIGING

It is still missing an “uncertainty map of the entire spatial domain”. The specification of the kriging method is given from line 781 to 799. Its validation procedure has been notably improved (Figure 2 and Table 3). However, being a probabilistic method, the kriging interpolation lets to have a measure of its uncertainty. In other words, it is possible to compute a measure of the deviation between the interpolated and the true values, say RMSE, averaged over the time domain. Then, map this measure throughout the entire spatial domain and not only for the monitoring station points. This map would be useful to visualize those areas where the interpolation method has lower performance and, consequently, to a better evaluation of the heatwave risk at a spatial level.

2) TREND

The sentence at line 1579 “linear regression with time series as independent variables and heatwave attributes (HWF, HWTD, and other 6 attributes) as dependent variables” could be substituted with “linear regression of heatwave attributes (HWF, HWTD, and other 6 attributes) on time”. Moreover, in line 1581, the text reports the significance for the statistical test at the level of “(P value > 0.05)” whereas it should be (P value < 0.05). Please, check that this is a typos error rather than a mistake in the computation of significative cases.

Author Response

Response to Reviewer 3 Comments

Thank you very much for your suggestions. The following reference line (e.g. Line 126) are based the version that all revisions are accepted.

Point 1: 1) KRIGING: It is still missing an “uncertainty map of the entire spatial domain”. The specification of the kriging method is given from line 781 to 799. Its validation procedure has been notably improved (Figure 2 and Table 3). However, being a probabilistic method, the kriging interpolation lets to have a measure of its uncertainty. In other words, it is possible to compute a measure of the deviation between the interpolated and the true values, say RMSE, averaged over the time domain. Then, map this measure throughout the entire spatial domain and not only for the monitoring station points. This map would be useful to visualize those areas where the interpolation method has lower performance and, consequently, to a better evaluation of the heatwave risk at a spatial level.

Response 1: Your suggestion is very constructive and points us to the steps. Based on your suggestion, we have added an uncertainty map for interpolation. The prediction variance map is an effective way to evaluate the effect of Kriging interpolation. The main point is: for each grid, calculate the variance of the grid point's multiple interpolation results. If the variance is smaller, it means that the result of each interpolation is closer. The more stable and reliable the results are. We calculated the prediction variance map of the entire time series, and obtained the final uncertainty map after averaging (Line 266).

Point 2: 2) TREND: The sentence at line 1579 “linear regression with time series as independent variables and heatwave attributes (HWF, HWTD, and other 6 attributes) as dependent variables” could be substituted with “linear regression of heatwave attributes (HWF, HWTD, and other 6 attributes) on time”. Moreover, in line 1581, the text reports the significance for the statistical test at the level of “(P value > 0.05)” whereas it should be (P value < 0.05). Please, check that this is a typos error rather than a mistake in the computation of significative cases.

Response 2: We modified this sentence based on your suggestions (Line 361). By checking the calculation results, p value> 0.05 is an error and we have revised it (Line 324 and 363).

This manuscript is a resubmission of an earlier submission. The following is a list of the peer review reports and author responses from that submission.

Round 1

Reviewer 1 Report

This is a manuscript in which the authors evaluate the spatio-temporal distribution of the heat wave risk assessment based on the apparent temperature.
From my point of view, the definition of the degree of risk definition in Table 2 does not take into account the impact on health.
The risk of temperature from the point of view of its impact on health varies from one place to another depending on multiple factors not only climatological such as demographic, economic and social factors, so that what can be a comfort time in one place it can be a risk of heat in another. There is multiple bibliography that supports this fact. The simplification made by the authors in Table 2 makes the results obtained meaningless from the point of view of human health risk.

Reviewer 2 Report

This manuscript (MS) examines the spatiotemporal distribution of heat waves based on apparent temperature in the One Belt and One Road (Silk Route) region and assesses the associated risk to people. Daily meteorological data 1989-2018 is used to calculate the daily apparent temperature dataset, annual heat waves and their spatiotemporal distribution, including neighboring regions. A 0.1°× 0.1° resolution is used and the daily apparent temperature is based on ambient temperature, relative humidity (RH) and wind strength. The changes in frequency and duration of heat waves were analyzed for the regions studied. It is concluded that that the heat wave risk in most areas along the One Belt and One Road route is relatively high and that the heat wave risk should be considered when considering processes for further developing the One Belt and One Road region.

(Just a comment, it would have been beneficial for the reviewer(s) if there were line numbers in the manuscript pdf.)

Title

The title is representative for the study.

Abstract

The Abstract describes what is done, is well written and it presents the main conclusions of the study.

General comments

Some "air" is missing before and after equations, figures, tables, bullet points, etc. Please confer the Remote Sensing Template and check this throughout the manuscript (MS).

Figure information is often not easily read.

Do not refer to figures, tables etc. in Bold. The Remote Sensing Template states: "All figures and tables should be cited in the main text as Figure 1, Table 1, etc." Check this throughout.

Introduction

Line 4, … changes in heat waves. ? (plural)

Page 2, Line 5: … a bout of intense heat waves … (plural)

Page 2, Line 15: … for disaster risk reduction, … ?  Check this also for other … disaster risk reduction

Defrance and Dimitri produced … ? Since Dimitri Defrance is the single author, should it rather not be: Defrance [38] produced …

Also check whether the reference numbers in correct sequence

Page 2 bullet point 3: … calculation of heat waves using … (plural)

Add an empty line before: Taking people's actual experience (temperature, … to show that this is not part of the bullet point 3.

The term "This study …", consider rather to use "The present study …" as this would help the reader to distinguish when referring to  your own study (the present study) or discussing somebody else’s study (his/her/their study). Check throughout.

The last bullet point at page 3 could maybe also include something about whether the findings could have any impact on the development of the OBOR region?

Materials and Methods

Section 2.1.

NOAA? Explain abbreviations first time they are used.

And, since you use so many abbreviations, should there rather be a list of abbreviations helping the readers when studying tables, figures, etc?

Line 6: Should there be a reference here: … processed using the random forest method [??].

Section 2.2.

Tc1, Ta and E are not explained after Equation (1). These must be explained.

Line 7: When introducing (0.0065 °C/m), use e.g. (E = 0.0065 °C/m) ? and explain the Tc1 and Ta immediately after Equation (1)?

Some "air" is missing before and after the line presenting Equation (1), please confer the Remote Sensing Template. Check this throughout.

2.2.2. Apparent temperature

The first sentence seems redundant since it is more or less repeated two sentences later.

The sentence … , which makes it more reliable to use the same threshold [??] throughout the study area. needs a reference so that the readers understand where this threshold comes from.

Humidex index [50]. The Humidex index [50] is increasingly…

Steadman index [53]. Steadman [53] published

(Else, check the Template and make these to become sub-subsections.)

With respect to Equations (4) and (5), is it needed to write , defined as Equation (4/5) in front of the equations? (Seems unnecessary, or?)

where H and A are apparent temperature (°C), Ta is the … Is both H and A apparent temperature, or is H the Humidity Index, as previously defined?

Ta is the ambient(?) temperature (°C),

E is the vapour pressure (pa), … Previously, E was defined as the temperature lapse rate in Equation (1)? The unit Pascal should be Pa, not pa.

2.2.3. Localized temperature threshold

Line 3, … temperature processes are called … processes? And, it would be best if this sentence includes a reference to the WHO heat wave definition.

In most parts of China, the standard for high temperature days is 35 °C [??], whereas in India, it is 40 °C [??]. These statements also would benefit from proper references.

Figure 3 and 4: Good figures, but the figure legends are so small that it is very difficult to read. The legends must be more readable.

Last sentence on page 9: The highest apparent temperature in each month generally appears in 2010, which indicates a severe heat wave in western Russia in 2010. Do you also have a separate reference for this heat wave in Russia?

Figure 5: a and b, make the x axis cross at y = -20 and -10, respectively. The y axis should be labeled "Apparent temperature (°C)", or "Ta (°C)", i.e. not simply Value. When improving the figure, try also to make all text on the figure easier to read (slightly larger font, maybe bold face, etc)

Figure 6, similar comments as for Figure 5.

Figure 7, 8 and 9: Similar comments as for Figure 3 and 4.

Page 14: Should the sentence: The end date of the last heat wave (Figure 9f). Most areas with continuous heat waves are positive areas, which indicates that the end date of the last heat wave has a delaying tendency. rather be a separate bullet point?

Page 15, 2nd bullet point: 73, 259 days for Dhaka ???

Figure 10: Not use Value on y axis, but rather something like "(Heat Wave) Frequency (y-1)"?

Figure 10: In the figure legend, (a), (b), (c), etc in bold

3.3. Heat wave risk assessment

Sentence 2: We first calculated the hazard, exposure and vulnerability of heat waves, and finally calculated the heat wave risk. Can this be rewritten so that we do not consider heat waves as vulnerable?

Bullet point 2: … which constitute the degree of exposure. ?  … which constitute the most risk exposed persons. ?

Bullet point 3: High-vulnerability areas are(?) located in areas

Discussion

The 1st sentence should preferably end up with a reference.

That Python was used to write the programs should preferably also have been mentioned earlier in the MS.

Raei, E. et al. calculated the … Rather: Raei et al. [30] calculated the … Check the references to Raei throughout as many/all(?) of these are incorrectly referenced.

Table 4: In the Table heading, refer to [30] if that is where the heat wave data based on ambient temperature are taken from. And, refer to the "present work" for the column with results from your study. Ansure proper "air" before and after Table 4, as is an issue several other places in the MS.

Where referring to … developed by li et al., and the …, rather use Li et al. [32]… Check throughout.

Table 5: If you use Area as the first column and Heat Wave Characteristic as the second column, I think the table becomes easier to read.

Table 6: Put reference numbers for each of the studies in the table.

Conclusions

Generally, this section is too long and should preferably be shortened significantly. A few specific comments:

3rd bullet point: Consider rewriting to tell that the heat waves occupy a larger part of the year, early as well as late in the year?

First of the last 5 bullet points: … avoid and mitigate heat wave disasters.

3rd of the last 5 bullet points: Do not introduce something that has not been treated earlier in the MS.

References

There are numerous deviations from the Remote Sensing Template. To name a few: write all names, and not "et al." in the list of references, no "and" between the names. Also include the DOI where available. You must carefully check the Template throughout.

Overall evaluation

Your present study was indeed an interesting read. In my opinion, it does however, need a major revision not based on major issues as such, but simply due to the numerous changes required throughout. I hope to receive an updated manuscript in the near future.

Reviewer 3 Report

General Comments
In this work the authors study the risk of heat waves between 1989 and 2018 in those countries that will participate in the economic project led by the Chinese government known as: “One Belt and One Road”. To this they use initially scattered data from weather stations that they later interpolate
in order to construct gridded maps. The authors define the risk of heat waves as the combination of heat wave hazard, exposure and vulnerability. To construct the former terms the authors use a suite of variables such as the population density or the distance to the closest body of water or hospital among others. General conclusions are that in summer the occurrence of heat waves is larger due to the stronger apparent temperatures, and that the frequency of heat waves is decreasing while their
duration is increasing with time leading to a major number of hot days. However, patterns present complex regional differences.
This work fits well in the journal and can be useful for the community of scientists working on this region. However, this paper needs major improvements before being considered for publication.
Key points to be improved are the structure, to write a better motivated story line, a clearer description of methods and a statement on data availability. Further details are shown below:

Major concerns

The structure of the paper needs to improve. There is a clear abuse of itemizations (In Section 1, 3-5) with results not well explained (only a list of short descriptions). Indeed I feel there is lack of connection in the story you are telling me, moving from one figure/table to another one without introducing the motivation for this.

For example, why do you think Fig. 7 and 11 are interesting?

Also, why do you prefer to show selected days (1st of each month) instead of seasonal averages (inFig. 1-7)? I do not see the point of showing selected days.
- Most of legends are hard to read (even with the pdf size in my laptop at 400%).

Sometimes numbers in legends are weird (e.g. Fig. 8). Please use natural numbers for frequency, simpler numbers for other cases, and make legends bigger.
- Regarding the followed methodology many points are unclear. In particular:

a) You define two indexes in Page 5 (Eq. 4 and Eq. 5), but then it is not clear at all which one are you using for plots in Fig. 8-13. Do you use each of them depending on the available data?
Indeed a comparison of both indexes should be shown in those locations where data are available.
Also complementary data used in Eq. 5 such as wind and relative humidity should be shown to help you (and the reader) to explain regional differences.

b) About the threshold selected to compute heat waves. As far as I understand, the authors first use Eq. 5 to compute the apparent temperature, interpolate it, and then you define a heat wave as an apparent temperature of at least 29ºC during 3 days. Am I right? Do you use the same definition of heat wave for the whole area?
c) The exact methodology to construct the hazard, exposure and vulnerability indexes (Eq. 6) is unclear to me. How do the authors combine the different variables to get this indexes and their classification (vey low, low, medium, high, very high)? What exact formula do you use?
- It is not mentioned in the paper but, will be this dataset publicly available? Otherwise I am not convinced about the actual usefulness of this work since nobody else can check or use it and there are no new explanations between regional differences.
- Spatial differences are not well discussed. It is Ok to me if the authors do not analyze in detail these differences but then they need to explain them citing previous works. It seems they have not included recent works such as Yu et al., (2019) or Mandal et al., (2019) or Russo et al., (2014), among others.

Other comments
Abstract
1) P. 1 Ln. 1. Suggest to change “additionally” by “therefore”. It looks a consequence of the first sentence.
2) P. 1 Ln. 4. Remove first “dataset” and change “of” by “for”.
3) P.1 Ln 11. Remove. “This indicates that the risk of heat waves is increasing”. It is redundant with respect to the previous sentence.
4) P. 1 Ln 11. Some rewriting: “The spatial distribution of the risk of heat waves …”
5) P.1 Ln 12. Change “The assessment result shows ...” by “Results show ...”
6) P.1 Ln. 15-17. I do not agree with this sentence. Fig. 12d does not show this, most of regions are low or medium risk.
Introduction
7) P.1 First paragraph. “… in heat waves. Model-based results show ...”
8) P. 1. Define an acronym for One Belt and One Road region to improve readability. May be OBOR?
9) P.1. “complex and diverse natural disasters”. Be more specific, give examples.
10) P. 1. Unclear what you mean by “ecological environment”. Do you refer to agriculture, biodiversity and natural resources?
11) P.2 first paragraph. Give more information on the duration of events when describing them.
12) P. 2 Referring works such us the first author surname + “team” must be avoided. Firstly because it may occur that the first author is a student and not the team leader. Use surname + number as you can see in other published articles in the journal.
13) P. 2. What does mean CEIs and CPC? Not specified.
14) P. 2, change GWHR by GHWR in items.
15) P. 2. Rewrite: “In adittion to temperature people actual experience is affected by other factors such as wind speed and relative humidity.”
16) P. 2 Change “of 1989-2018” by “for 1989-2018”
17) P.3. Ln 1. “… heat waves maximum ...” Materials and Methods
18) P.3 “… random forest method”. Add some reference here on this method.
19) P.3 The authors say: “…, the night-time lights data represent the population distribution and the level of economic development.” I am not sure that the second part is true. There are rich and poor very populated cities. Do you have a reference for this?
20) P.3 I think section 2.1 would benefit from a better motivation of the required data to estimate the risk of a heat wave. For example by describing data for apparent temperature and then Fig. 2 here, and introducing briefly the concepts of hazard, exposure and vulnerability.
21) P.4 Why in Fig. 1c Poland does not have hospitals?
22) P.4 Legends in Fig. 1 (and in some other) are difficult to read. Also the short colorbar is not very useful.
23) P.4 “First, an initial correction of temperature ...”.
24) P.4 After Eq. (1). “Then, the corrected temperature is interpolated to a synthetic mesh of 0.1º x 0.1º”.
25) P.4 Change “the” by “a” in the last line.
26) P.5 “On the assumption of a zero plane”.
27) P. 5 Use the same notation for apparent temperature (Eq. 4 and Eq. 5). Indeed E is a repeated letter previously used for elevation in Eq. 3.
28) P. 6 first line. Here you say that a temperature threshold was defined according to different percentiles. Later you say that a heat wave is defined when apparent temperature is larger than 29ºC. Unclear to me what exactly you are doing here.
29) P. 6 “… by a heat wave event”.
Results
30) P. 8. Fig. 3. Please simplify captions to improve readability, e.g. “Spatial distribution of mean apparent temperature between 1989-2018 for: a) January 1st, b) March 1st, c) May 1st, d) July 1st, e) September 1st and f) November 1st.” Check all other captions.
31) P. 8. You refer to a “linear” trend right. Specify it.
32) P.8 Difficult to see if there is a trend in Fig. 4 as they seem to be compensated seasonaly. Also makes more sense to me to use seasonal averages as I suggested above.
33) P. 10 Why have you chosen to show these cities in Fig. 5. Please motivate more this election.
34) P. 10 In Fig. 5 you are showing the daily maximum, average or minimum temperature. It is not clear from the caption.
35) P. 10 Suggest to move section 3.1.3 to section 2.
36) P. 10 In Section 3.1.3, in the third item. I assume you want to say: “The apparent temperature at the training set stations ...” not validation.
37) P. 11 Show values of slopes in Fig. 6.
38) P. 12 Fig. 7. Why year 2018?
39) P. 12 “29ºC is the considered to be the limit of comfortable apparent temperature.” This is different from Table 2, in which comfort is defined between 18ºC and 23ºC. Please be consistent.
40) P.13 Fig. 8b indicates that in Indonesia, Malaysia and other regions there is almost permanent heat wave (1 per year that lasts more than 300 days). Does it make any sense from an extreme event perspective? Also use a more readable colorbar range in legends.
41) P. 14. Fig. 9d. “… most areas are positive ...”. Difficult to say if this is true.
42) P. 15. Ln 2 and 5. Remove extra-numbers.
43) P. 15 Show slopes in Fig. 10.
44) P.16. First two lines. Have you made a sensitivity analysis? What happens if the threshold is modified? Do results change significantly?
Discussion
45) P.18-19. I think Section 4.1 should be moved to Section 2.
46) P. 20. Unclear which colorbar range are you using for Fig. 13 top plots.
47) As said above, the discussion can be improved by adding a more complete discussion on the reasons for regional differences (closer regions in Fig. 12 show different patterns), with a sensitivity analysis and by adding the comparision with more recent omitted results. Conclusions
48) P. 22. Areas of high exposure are very difficult to see in Fig. 12.
49) Page 22. Suggestions are moslty common sense, not a result of your work. I would not include them as they are too general and not new.
50) Please write well structured paragraphs and not a set of itemized sentences without a logic link between them.
References
51) Change names by surnames in reference 51.